# Tumor-reprogrammed resident T cells resist radiation to control tumors

Ainhoa Arina [1], Michael Beckett[1], Christian Fernandez [1], Wenxin Zheng[1], Sean Pitroda [1], Steven J. Chmura[1], Jason J. Luke[2], Martin Forde[1], Yuzhu Hou[1], Byron Burnette[1], Helena Mauceri[1], Israel Lowy[3], Tasha Sims[3], Nikolai Khodarev[1], Yang-Xin Fu [4] & Ralph R. Weichselbaum[1]

Successful combinations of radiotherapy and immunotherapy depend on the presence of live T cells within the tumor; however, radiotherapy is believed to damage T cells. Here, based on longitudinal *in vivo* imaging and functional analysis, we report that a large proportion of T cells survive clinically relevant doses of radiation and show increased motility, and higher production of interferon gamma, compared with T cells from unirradiated tumors. Irradiated intratumoral T cells can mediate tumor control without newly-infiltrating T cells. Transcriptomic analysis suggests T cell reprogramming in the tumor microenvironment and similarities with tissue-resident memory T cells, which are more radio-resistant than circulating/lymphoid tissue T cells. TGFβ is a key upstream regulator of T cell reprogramming and contributes to intratumoral Tcell radio-resistance. These findings have implications for the design of radio-immunotherapy trials in that local irradiation is not inherently immunosuppressive, and irradiation of multiple tumors might optimize systemic effects of radiotherapy.

[1] Department of Radiation and Cellular Oncology and Ludwig Center for Metastasis Research, The University of Chicago, Chicago, IL 60637, USA. [2] Department of Medicine, The University of Chicago, Chicago, IL 60637, USA. [3] Regeneron Pharmaceuticals, Tarrytown, NY, USA. [4] Department of Pathology, University of Texas Southwestern, Dallas, TX 75390, USA. Correspondence and requests for materials should be addressed to A. A. (email: aarina@bsd.uchicago.edu) or to R.R.W. (email: rrw@radonc.uchicago.edu)

Lymphocytes have long been considered the most radio-sensitive cells in the mammalian body[1], based on seminal studies from the 1930 to 1950s, which estimated the dose required to kill 50% of the cells as 150 cGy for lymph node lymphocytes, compared with $10^3$–$10^5$ cGy for other non-mitotic cells[1,2]. Nuclear degeneration is the earliest visible change and takes place between 1 and 6 h after irradiation of lymphocytes[1,3,4]. Ionizing radiation (IR) induces interphase death[5] via p53-dependent apoptosis in TCRα/β$^+$ lymphocytes[6] following DNA damage. Not all T cell subsets are equally radiosensitive, e.g. CD4$^+$ regulatory T cells, antigen-experienced, and memory T cells have been all shown to be more radioresistant than naïve T cells[7–10], and circulating CD8$^+$ T cells have been defined by some studies as the most radiosensitive population[11,12]. As a consequence, whole-body irradiation (WBI) results in a marked reduction in primary immune responses to antigen[5]. Therefore, exposure to IR has been commonly considered as highly immunosuppressive[13]. In apparent contrast, increasing preclinical evidence from our group and others indicates that IR can also potentiate immune responses to tumors, and T cells are in many cases required for IR to exert its full antitumor effect (reviewed in refs. [14,15]). Multiple mechanisms exist by which IR can promote antitumor T cell immunity[14–16], including the sensitization of cancer cells to T cell killing by MHC-I and Fas up-regulation, increased antigen availability, and an improved antigen presenting function resulting from the release of damage signals and pro-inflammatory cytokines such as type I interferon[17]. IR can also increase T cell infiltration[18–20] by inducing T cell-recruiting chemokines[19,21] and vascular normalization[22]. This increase in T cell infiltration, together with the high radiosensitivity of lymphocytes as discussed above, may lead to the notion that T cell-mediated effects of IR depend solely on the recruitment of new T cells into the tumor.

The important role played by local immunity has recently been unveiled by studies on tissue-resident T cells and viral infection. Tissue-resident memory cells ($T_{RM}$) are far superior to circulating memory cells in reducing viral load upon re-challenge[23–25] even in the absence of new infiltrating T cells[24]. In cancer patients, the abundance of T cells with a $T_{RM}$ phenotype correlates with longer disease-free and overall survival in multiple cancer types (reviewed in ref. [26]), and is in some studies a better prognostic factor than the total number of CD8$^+$ T cells[27,28]. Antitumor vaccines were more effective when they induced $T_{RM}$ in addition to systemic immunity[29], all of which points at an important role of $T_{RM}$ in antitumor responses. While all $T_{RM}$ share a core gene expression signature[30,31], $T_{RM}$ from different locations exhibit tissue-specific adaptations at the transcriptional level[31,32]. In the local tumor microenvironment, cell-to-cell/cell-to-extracellular matrix interactions or soluble factors may provide unique pro-survival signals to T cells, raising the possibility that preexisting intratumoral T cells could have a contribution in the immune-mediated effects of IR.

Combining immunotherapy with radiotherapy has garnered much interest recently, and over 200 clinical trials are currently investigating immunotherapy and radiotherapy combinations. Here, we investigate the role of preexisting tumor-infiltrating T cells in the immune responses induced by IR against tumors. We find that, in murine tumor models representative of "inflamed" human tumors (i.e. with preexistent tumor-infiltrating lymphocytes), many of these preexistent T cells not only survive even high doses (20 Gy) of localized IR but also show an improved effector function 9 days after IR and mediate tumor growth control without the contribution of newly infiltrating T cells. Transcriptomic analysis of intratumoral T cells shows genetic reprogramming by the tumor microenvironment and a significant overlap with gene expression patterns of non-lymphoid tissue-resident T cells, which are also radio-resistant. Our results open new pathways for optimization of radio/immunotherapy combinations by uncovering a role for preexistent intratumoral T cells in the therapeutic response to IR.

## Results

**Intratumoral T cells survive following local IR treatment.** To quantify new infiltration vs. survival of preexisting T cells within tumors after IR, we performed in vivo longitudinal tumor imaging experiments in mice bearing implanted dorsal window chambers[33,34], in which preexisting T cells and newly infiltrating T cells were differentially labeled (Fig. 1a, b). For these experiments we used the Panc02 pancreatic cancer cell line transduced with a trackable model antigen (Panc02SIYCerulean)[20]. A single 200 μg dose of anti-CD8 depleting antibodies on day 0 allowed SIY-expressing tumors grow more aggressively and in all mice (Supplementary Fig. 1A) by transiently eliminating circulating and intratumoral CD8$^+$ T cells (Supplementary Fig. 1B). By the time tumors were established and imaging experiments were started, CD8$^+$ T cells had recovered and infiltrated tumors abundantly, and expressed markers associated with functional exhaustion (Supplementary Fig. 1C). CD4$^+$ T cells were also present, and 27% were FoxP3$^+$ regulatory T cells (Supplementary Fig. 1C). Thus, our model represents immunogenic/"inflamed" tumors with abundant T cell infiltration that fails to eradicate the tumor.

To eliminate circulating/peripheral T cells, mice with established tumors were treated with a myelo-ablative (8 Gy) dose of WBI. Tumors in the window chambers were shielded from WBI using lead to preserve EYFP$^+$ intratumoral T cells (Fig. 1c). Bone marrow was reconstituted with DsRed$^+$Rag$^{-/-}$ cells. Then mice were injected with in vitro-activated EGFP$^+$ 2C transgenic T cells specific for the SIY antigen, to track new T cell infiltration. 2C$^+$ EGFP$^+$ T cells became visible in the tumor 3–4 days after transfer (Fig. 1d). At that time, one mouse in each experiment was treated with local IR, while the second (control) mouse was untreated. Two IR protocols relevant to clinical practice were tested in independent experiments, one modeling fractionated IR (5 doses of 1.8 Gy separated by 24 h) and the other modeling Stereotactic Body Radiotherapy (SBRT, 20 Gy single dose). Figure 1c shows that after either fractionated IR or SBRT-like doses, a substantial fraction of preexisting EYFP$^+$ T cells were preserved for at least 9–14 days post-IR (85% and 65% of the initial pre-IR average EYFP$^+$ T cell counts, respectively, in the last measured time point). At the time of local IR, the number of EYFP$^+$ T cells in the circulation stayed at less than 10% of the pre-WBI levels (Supplementary Fig. 2); therefore, it is unlikely that peripheral EYFP$^+$ T cells surviving WBI would contribute significantly to the number of EYFP$^+$ quantified in tumors after IR. Peripheral EGFP$^+$ newly infiltrating T cells experienced a slight delay in infiltration in both mice receiving local IR, but eventually reached maximum numbers similar to those in non-irradiated mice (Fig. 1d). Phenotypic analysis of differentially labeled preexistent and newly infiltrating T cells revealed that the majority of cells in both populations were CD44$^+$CD62L$^-$ (Supplementary Fig. 3A, B). Preexisting T cells showed a comparatively lower Ki67 staining (Supplementary Fig. 3C), suggesting a slower proliferation compared with newly infiltrating T cells. Preexisting intratumoral T cells also had higher levels of PD1 and CD39 surface markers than newly infiltrating T cells (Supplementary Fig. 3D, E), consistent with a more exhausted phenotype or differences between a polyclonal (preexistent) vs. monoclonal (new) T cell population. These differences became even more pronounced after IR (Supplementary Fig. 3E). Strong gamma-H2AX staining at 1 h (Supplementary Fig. 3F) confirmed DNA damage. To

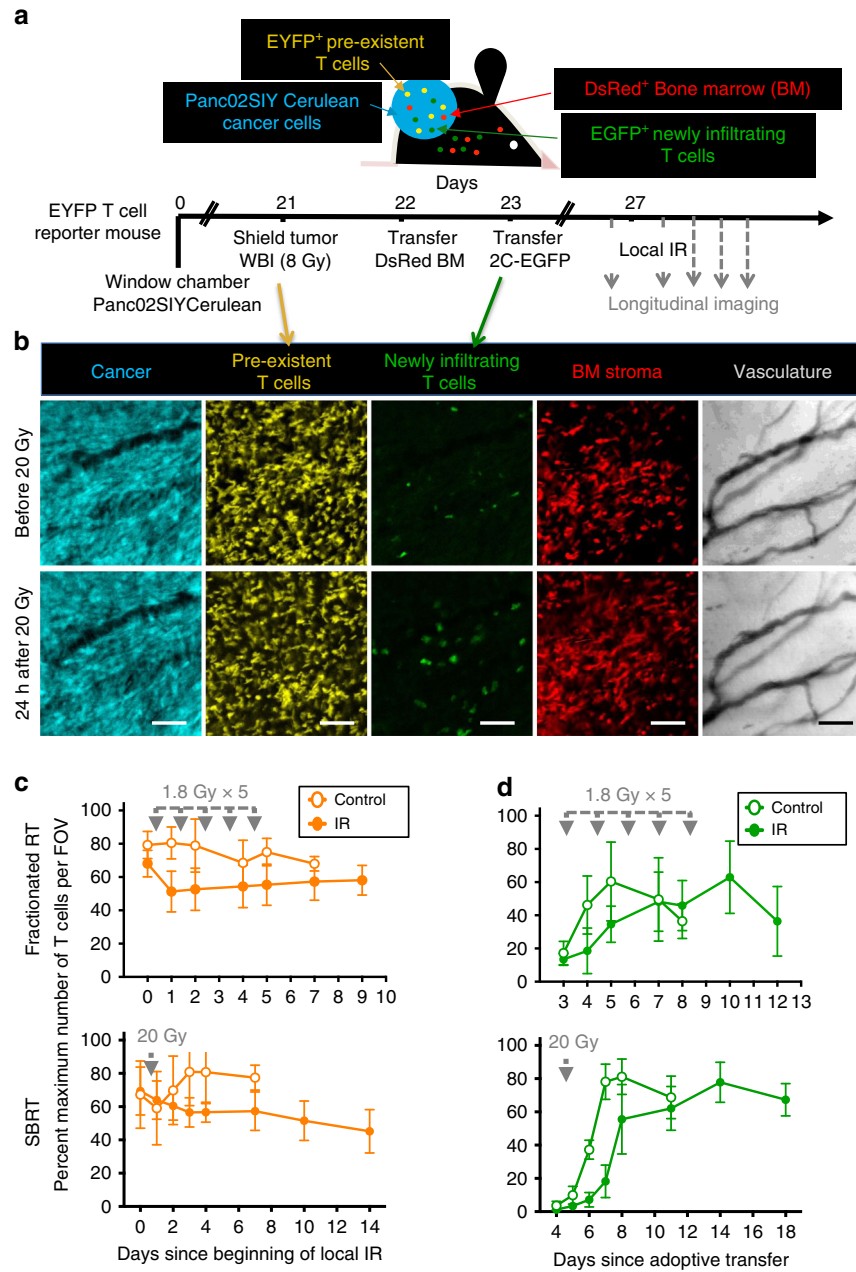

**Fig. 1** New as well as IR-resistant preexisting T cells contribute to T cells found in irradiated tumors. **a** Experimental design for longitudinal imaging of locally irradiated tumors. Panc02SIYCerulean cancer cells were injected s.c. into T cell reporter (Lck-EYFP) mice bearing dorsal window chambers. On day 21, mice received 8 Gy WBI while tumors were shielded, to deplete peripheral T cells and preserve preexistent intratumoral EYFP+ T cells. Following bone marrow reconstitution with DsRed+Rag−/− cells, EGFP+ 2C CD8+ T cells specific for the SIY tumor antigen were adoptively transferred. Three to four days after 2C transfer, mice received different treatment schedules of IR or no local IR (control mice). **b** Images are from a representative tumor region before and 24 h after treatment with 20 Gy (×20, scale bar: 100 μm). **c**, **d** Experiments were performed using two IR schedules modeling fractionated IR (5 doses, 1.8 Gy each) and SBRT (a single dose of 20 Gy). Preexisting EYFP+ (**c**) and newly infiltrating EGFP+ (**d**) T cell numbers were determined over time in multiple optical regions per mouse and normalized to the maximum count observed for each T cell type (averages and SD shown). Number of optical regions (N) was as follows (time after IR; N in control mouse, N in IR mouse): (d0; 7, 8–d1; 7, 7–d2; 7, 9–d4; 9, 10–d5; 9, 10–d7; 6, 10–d9; NA, 9) for the 1.8 Gy × 5 experiment, and (d0; 11, 19–d1; 10, 19–d2; 10, 19–d3; 10, 20–d4; 7, 21–d7; 10, 7–d10; NA,11–d14; NA, 9) for the 20 Gy × 1 experiment. The average EYFP (and EGFP) counts over time were positive with a 95% confidence level using quadratic or linear regression models, thus proving that IR did not deplete T cells. Data are representative of two independent longitudinal experiments performed for each treatment modality

extend the findings on intratumoral T cell survival after IR, a second tumor model and higher IR dose were used. T cell reporter mice bearing MC38 tumors were treated with a total dose of 30 Gy (10 + 20 Gy separated by 4 days) or no local IR (Supplementary Fig. 4). The first 10 Gy dose caused the largest reduction in T cell numbers. However, at all time points,

including those obtained after the 20 Gy dose, preexisting EYFP+ T cells were detectable.

Effector T cells actively scan peripheral tissues in search for their targets[35]. T cell motility in tumors is often compromised[35] and IR can increase the motility of infiltrating T cells[36]. To determine the viability and functionality of intratumoral T cells

exposed to IR, the motility of these cells was analyzed before and after IR in Panc02SIYCerulean tumors. As an unirradiated control, the motility of newly infiltrating T cells present in the same tumor regions was analyzed. The motility of preexisting EYFP+ T cells did not decrease after 20 or 1.8 Gy IR, but increased to a similar extent ($P < 0.0001$ for average speed and arrest coefficient changes between day 0 and 1 for both doses), as expected from live, viable cells. Furthermore, preexisting T cell average speeds and arrest coefficients were comparable to those of newly infiltrating T cells at all time points tested (Fig. 2, Supplementary Movies 1 and 2). Therefore, live preexisting intratumoral T cells were detected for at least 1–2 weeks after IR.

**Organ-dependent radiosensitivity of T cells within tissues**. The persistence of viable intratumoral T cells after IR suggested a relatively radio-resistant phenotype. We quantified the number of T cells in the tumor and peripheral blood of MC38 tumor-bearing mice that received increasing doses of WBI 24 h prior to sacrifice. WBI resulted in a dose-dependent loss of circulating but not tumor-infiltrating T cells (Fig. 3a, Supplementary Fig. 5). The majority of T cells in tumors at any WBI dose were CD8+ T cells, in contrast with circulating CD8+ T cells, whose percentage dropped with increasing doses of WBI (Fig. 3b). To determine whether a solid organ local microenvironment mediated tumor T cell radio-resistance, we quantified the numbers of CD8+ T cells in several solid organs after WBI of mice with a single dose of 8 Gy (Fig. 3c, P values in Supplementary Table 1). Intravascular staining[37] was used to exclude enumeration of cells present in the microvasculature. We found different degrees of radiosensitivity among parenchymal CD8+ T cells from different organs. Lymphoid organs (lymph nodes and spleens) had the most radiosensitive CD8+ T cells, whereas CD8+ T cells in the gut (IEL, intraepithelial lymphocytes) and the tumor were the most radioresistant, with no significant differences between the two in terms of IR-induced decrease in T cells per gram of tissue. Liver T cells showed intermediate radiosensitivity. We hypothesized that the higher radio-resistance of parenchymal CD8+ T cells from non-lymphoid compared with lymphoid solid organs could be explained by the presence of $T_{RM}$. We found that higher percents of $T_{RM}$ (defined as CD69+CD103+ in IEL and tumor, and CD69+LFA1+ in liver) (Supplementary Fig. 6A, B) associated with smaller effects of IR in the numbers of parenchymal CD8+ T cells (Fig. 3d, Supplementary Fig. 6C). Liver CD8+ T cells within the tissue had the lowest % $T_{RM}$ at baseline, but this % $T_{RM}$ increased significantly after IR, suggesting that $T_{RM}$ constituted the radioresistant CD8+ T cell subpopulation in the liver (Fig. 3e). % $T_{RM}$ in IEL and tumor showed a trend to increase after IR, but the difference did not reach statistical significance. To directly test a differential radiosensitivity between $T_{RM}$ and the rest of CD8+ T cells within non-lymphoid solid organs and tissues, we calculated the effect of IR separately in these two subpopulations. As can be seen in Fig. 3f, the effect of IR was indeed lower in $T_{RM}$ compared with non-$T_{RM}$ in the liver; however, for IEL and tumors, the sensitivity of $T_{RM}$ and non-$T_{RM}$ CD8+ T cells was similar. These results suggest that $T_{RM}$ within certain solid organs have a higher radio-resistance compared with T cells in circulation and in lymphoid organs; however, in tissues harboring the most radioresistant CD8+ T cells (IEL and tumor) not only cells with the standard $T_{RM}$ phenotype but all CD8+ T cells were similarly radioresistant. CD8+ T cells within lymph node tissue included a small proportion of CD44−CD62L+ naïve T cells (more radiosensitive than memory T cells[9]), but in the other organs/tissues, the percent of naïve cells was even smaller (Supplementary Fig. 6D), and did not correlate with T cell radiosensitivity, e.g. spleen and IEL T cells had similar

percentages of naïve T cells, but very different radiosensitivity, as shown in Fig. 3c. Differences in proliferation in the steady state could not explain the different radiosensitivity of T cells from lymphoid versus non-lymphoid tissues and tumor either, since Ki67 levels were highest for intratumoral parenchymal CD8+ T cells and similar for lymphoid and non-lymphoid tissue-parenchymal CD8+ T cells (Supplementary Fig. 7), and thus did not correlate with sensitivity to IR. Overall, different organ environments and parenchymal CD8+ T cell phenotypes, but not proliferative status, were associated with different degrees of T cell radiosensitivity in solid organs.

**Tumor microenvironment-mediated T cell reprogramming**. Based on the observed organ-dependent sensitivity to IR, we hypothesized that the tumor microenvironment transcriptionally reprograms tumor-infiltrating T cells, contributing to their radioresistance[38,39]. We analyzed the transcriptome of CD8+CD44+ CD62L− T cells purified from MC38 tumors and from lymph nodes (LN) of irradiated and control mice (Supplementary Fig. 8, Fig. 4a), 5 h after WBI. T cells from lymph nodes were selected for comparison because they are most similar to tumor T cells in many aspects other than their sensitivity to IR. Like tumor T cells, lymph node T cells are in a solid organ, and they are presumably the source of antigen-primed T cells that then traffic to the tumor and eventually become tumor-residing T cells. CD8+CD44+ CD62L− T cells were specifically sorted because that was the phenotype of most MC38 tumor-infiltrating CD8+ T cells (Supplementary Fig. 6D). Despite the difference in magnitude in gene response to IR (30 vs. 447 genes) (Fig. 4b), both LN and tumor T cells responded to IR primarily activating p53 signaling (Supplementary Fig. 9), as expected. Consistent with previous findings[40], basal gene expression differences between tissues of origin (2454 genes) (Fig. 4b, c) were much more numerous compared with IR-induced transcriptional changes.

We next analyzed baseline differences in gene expression between tumor and LN-derived T cells as potential explanations for the differential radiosensitivity observed. Gene set enrichment analysis revealed that the most significantly enriched hallmark signatures in intratumoral T cells corresponded to angiogenesis and epithelial–mesenchymal transition (EMT) (Supplementary Table 2). Many over-expressed genes in tumor-associated T cells were involved in tissue remodeling (e.g., *Col1a1, Col4a1, Fn1, Lamc1, Mmp2, Mmp10, Timp3*) and cell motility/adhesion (*Rhoc, Itga1, Itgav*) (Supplementary Fig. 10). IPA, GSEA, and Gene Ontology analysis identified in tumor T cell-associated differentially expressed genes (DEG), signatures related with cancer, ECM remodeling, invasion, motility and adhesion (Supplementary Fig. 11). Flow cytometry confirmed the up-regulation of integrin subunits α1, αV, αM, and α9 at protein level (Supplementary Fig. 12), which validated RNA array data. Functional analysis detected TGFβ as the top upstream regulator of T cell reprogramming in the tumor microenvironment (Supplementary Data 1). Overall, tumor-infiltrating T cells were reprogrammed by the tumor microenvironment to express angiogenic and tissue-remodeling pathways more typical of cancer than immune cells.

**Intratumoral T cells and $T_{RM}$ share a similar transcriptome**. Since T cells within tumors and $T_{RM}$ from solid, non-lymphoid organs shared some degree of radio-resistance, and gene expression data pointed to TGFβ, which is necessary for $T_{RM}$ development[30,41], as one key upstream regulator of intratumoral T cell reprogramming, we hypothesized that intratumoral T cells and $T_{RM}$ share similarities at the transcriptional level. We compared our array data with published datasets of $T_{RM}$ vs. spleen memory T cell subsets in HSV-infected mice[30]. We found that

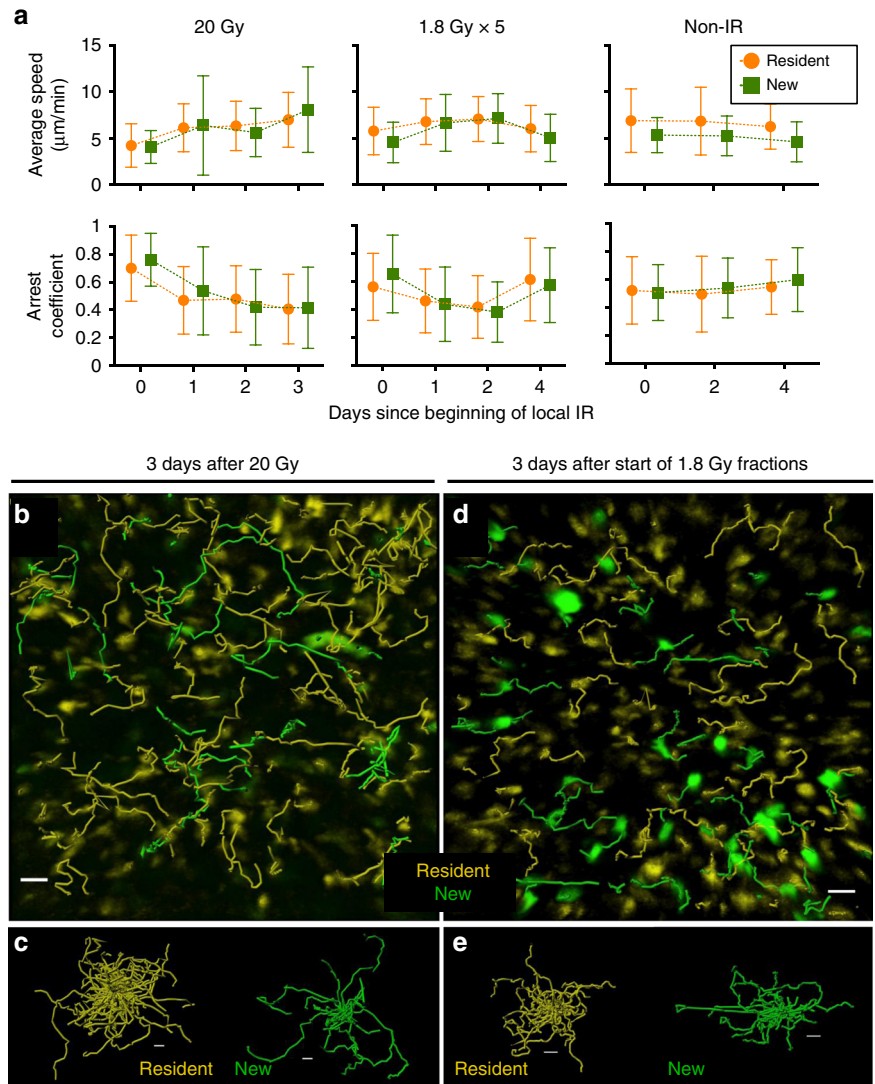

**Fig. 2** Preexistent intratumoral T cells maintain their motility after IR. T cell motility was analyzed in the experiments from Fig. 1. **a** The motility of irradiated preexisting EYFP[+] cells is comparable to newly infiltrating T cells present in the same tumor regions, as evidenced by similar average speeds and arrest coefficients at any time point analyzed. Data are from 20 to 25 min movies and 21 to 35 μm deep optical sections (Z-stack). Comparisons for average speed and arrest coefficient between preexistent and new T cells were non-significant at all time points post-IR (unpaired $t$-test). The number (N) of cell tracks quantified was as follows (day after beginning of IR; N preexistent T cells, N new T cells): 20 Gy mouse (d0; 204, 10–d1; 216, 18–d2; 184, 40–d3; 114, 33), 1.8 Gy × 5 mouse (d0; 471, 56–d1; 238, 62–d2; 149, 51–d4; 232, 53), Non-IR (d0; 206, 36–d2; 207, 86–d4; 254,104). **b**, **d** Tracks followed by preexistent (yellow) and new (green) T cells within the same tumor region 3 days after IR. **c**, **e** Spider plots show separately preexistent and new T cell tracks, from **b** and **d**, respectively. Scale bars: 50 μm

our tumor T cell samples clustered with skin and lung $T_{RM}$ samples, whereas our LN T cell samples clustered with all types of spleen samples from the previously published datasets (Fig. 5a). Eighteen out of 37 genes from the core signature of $T_{RM}$ cells from different organs[30] overlapped with our tumor-associated T cell signature (Fig. 5b). These data further demonstrated the similarities between $T_{RM}$ and intratumoral T cells.

**Anti-TGFβ partially radiosensitizes intratumoral T cells.** Based on our transcriptional findings, we next tested a possible role for TGFβ in the radio-resistance of tumor-infiltrating T cells. Antibody-mediated TGFβ blockade increased the number of T cells in tumors in the absence of IR, consistent with published data[42,43] (Fig. 5c), and resulted in a significantly smaller fraction of T cells (both CD8[+] and CD4[+]; Supplementary Fig. 13) surviving WBI in tumors, compared with IgG-treated animals

(Fig. 5d). These data suggest that TGFβ present in the tumor microenvironment might contribute to the radio-resistance of intratumoral T cells.

**Preexisting T cells can mediate the antitumor effects of IR.** We evaluated the functional consequences of IR exposure on intratumoral T cells. MC38 tumor fragments containing EYFP[+] T cells from mice treated with increasing doses of WBI (as in Fig. 3) were injected s.c. in OT1Rag[−/−] recipients to assess the long-term survival of irradiated intratumoral T cells in the new hosts. Three weeks after tumor fragment implantation, EYFP[+] T cells had expanded in the new recipient mice and could be detected in the spleen in increased numbers in mice receiving no/low-dose IR compared with high-dose IR-treated fragments (Fig. 6a, Supplementary Fig. 14). These results suggested that either the T cells receiving high-dose IR had died after transfer but prior to day 21 or

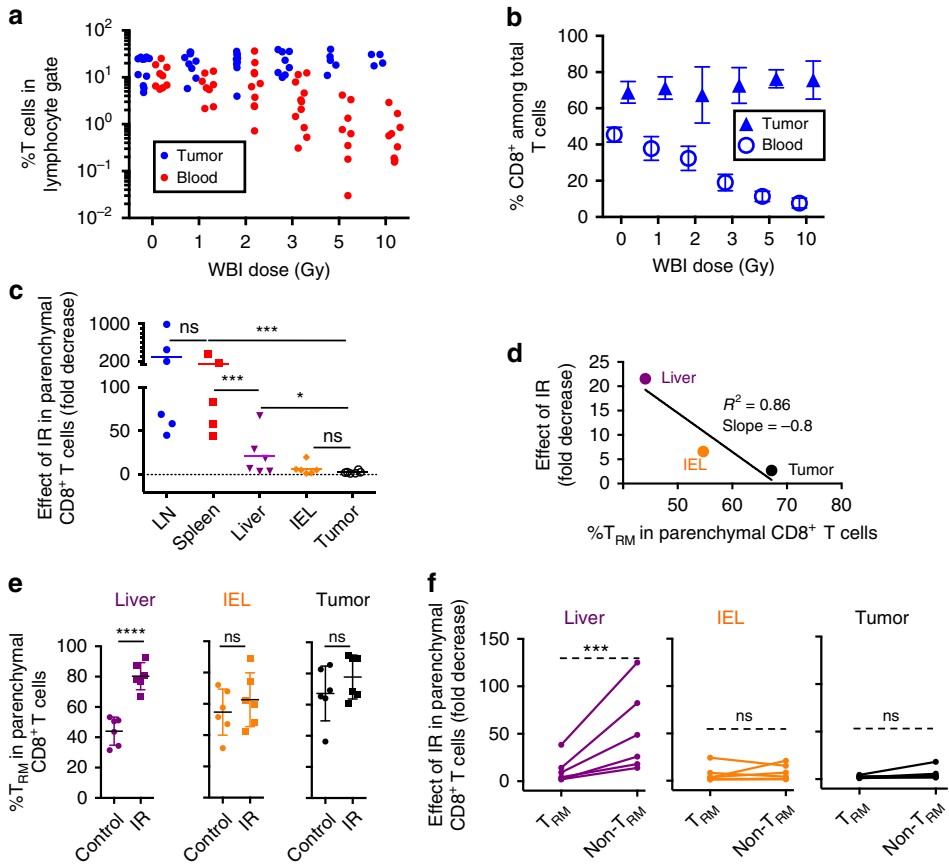

**Fig. 3** Increased radio-resistance of T cells within tumors and non-lymphoid solid organs compared with circulating T cells and T cells from lymphoid organs. **a** EYFP[+] T cell reporter mice bearing established MC38 tumors received different doses of WBI, as indicated. Twenty-four hours later, survival of T cells in tumors vs. circulation was determined by flow cytometry (gating in Supplementary Fig. 5). The tumor/blood slopes were significantly different ($P$ = 0.02) by linear regression analysis. **b** Percentage of CD8[+] cells among total tumor and blood T cells from the experiment shown in **a**. % CD8[+] cells in blood was significantly lower after all WBI doses tested compared to unirradiated mice ($P$ = 0.01 at 1 Gy, <0.001 for the rest, unpaired $t$-test). In tumors, differences were not significant or % CD8[+] T cells was higher in irradiated compared to unirradiated mice ($P$ = 0.03 at the 5 Gy dose). Data in **a** and **b** were pooled from three independent experiments comparing tumor and blood and one additional experiment with data only from blood. In **a**, each dot represents an individual mouse whereas **b** shows average and SD. **c–f** C57BL/6 mice bearing established (3 weeks) MC38 tumors were treated with 8 Gy WBI (IR) or left untreated (control). Twenty-four hours later, absolute numbers of parenchymal (i.v. antibody-negative) CD8[+] T cells per gram were determined for the organs and tissues indicated, and fold decrease was calculated in each irradiated mouse and tissue relative to the average number of T cells per gram of tissue in unirradiated mice in the same experiment. Data are pooled from two independent experiments, $N$ = 6 total mice per organ and condition. **c** shows averages, $P$ values by ratio $t$-test. Exact $P$ values in Supplementary Table 1. **d** Average fold decrease in parenchymal CD8[+] T cells was plotted against the %$T_{RM}$ in each non-lymphoid solid organ and analyzed by linear regression. **e** Percent of cells with $T_{RM}$ phenotype (CD103[+]CD69[+] for IEL, tumor, and CD69[+]LFA1[+] for liver) in each organ in mice that received or not IR (average and SD, $P$ values by unpaired $t$-test). **f** Effect of IR on $T_{RM}$ vs. non-$T_{RM}$ CD8[+] cells (ratio-paired $t$-test). n.s., non-significant; ***$P \le 0.001$; ****$P \le 0.0001$

they failed to homeostatically proliferate in the recipient mice. To test the proliferative ability of T cells after IR, we isolated CD8[+] T cells from MC38 tumor-bearing mice that had received 5 Gy WBI or no IR 24 h earlier, and labeled them with CFSE. After 3 days of in vitro stimulation, non-IR T cells had diluted CFSE to a much greater extent than 5 Gy-treated T cells, indicating that irradiated T cells had indeed lost their ability to divide and were growth-arrested (Fig. 6b). To determine whether irradiation had affected T cell effector function, we analyzed the ability of T cells to produce IFNγ from tumors treated locally with 20 Gy. To limit the functional analysis to irradiated preexistent T cells, excluding new infiltration, we treated the mice with FTY720 every 24 h starting one day before IR (Fig. 6c). FTY720 blocks T cell egress from lymphoid tissue[44], resulting in the disappearance of T cells from circulation (Supplementary Fig. 15) for the duration of treatment. Surprisingly, intratumoral T cells isolated from tumors 9 days post 20-Gy treatment produced higher amounts of IFNγ than those from non-irradiated tumors (Fig. 6c). Most (85%) of the increase in

IFNγ production caused by IR was still observed in the FTY720-treated group compared with non-FTY720-treated animals, suggesting preexisting T cells may account for most of the IFNγ production in response to IR (Supplementary Fig. 16). A time-course experiment testing the function of CD8[+] intratumoral T cells isolated from tumors treated with 20 Gy showed that increased T cell function is not detected immediately after IR, but 5–9 days later (Supplementary Fig. 17A).

Tumor irradiation resulted in a decreased suppressive function of total CD11b[+] cells between 3 and 5 days post-IR (Supplementary Fig. 17B, C), which coincided with small changes in the composition of CD11b[+] cells (Supplementary Fig. 17D, E). These changes were most significant at day 5 after IR and consisted of increased percentages of CD11b[+]Ly6C[hi] monocytic and CD11b[+]Ly6G[+] granulocytic cells and decreased percentages of macrophages among CD11b[+] cells. Antigen-presenting function of total CD11c[+] cells isolated from irradiated Panc02SIY tumors seemed to be highest at day 5, although the difference with unirradiated

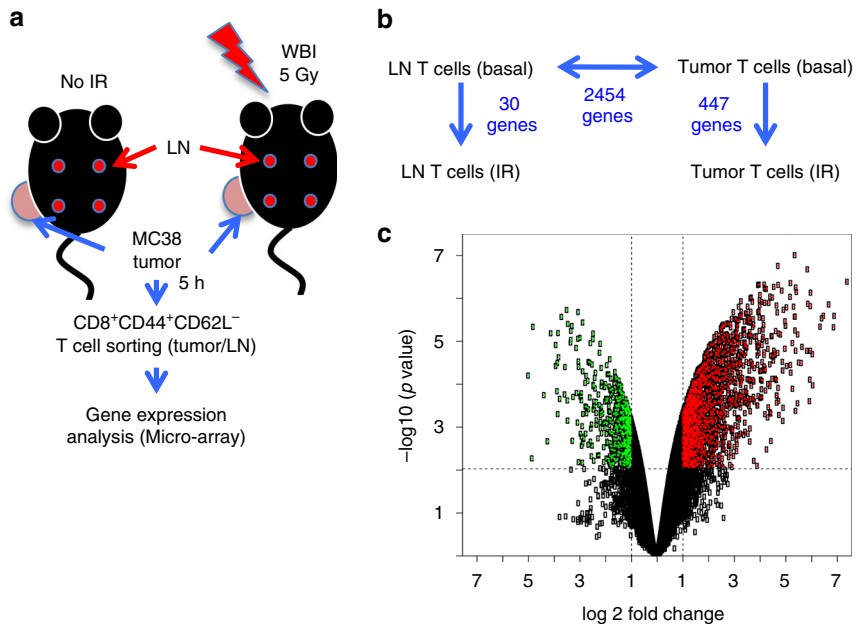

**Fig. 4** Transcriptional reprogramming of T cells in the tumor microenvironment. **a** Experimental scheme. **b** Overview of differential gene expression between the four groups compared. **c** Volcano plot showing the overall number of genes up- (red) or down-regulated (green) in tumor T cells compared with LN T cells (basal, i.e. unirradiated mice), and the magnitude of the change

mice did not reach statistical significance (Supplementary Fig. 18A). At that time point (day 5), the % of DCs was increased within CD45[+] cells isolated from tumors (Supplementary Fig. 18B). The CD11b[+] but not CD103[+] DC subset was enriched after IR and showed higher expression of MHCII, indicating that CD11b[+] might be the DC subset most involved in the response to IR (Supplementary Fig. 18C, D). Taken together, these results suggest that there is a window of time during which IR-induced changes in the tumor myeloid cell composition precede the maximum functional reactivation of irradiated preexisting intratumoral T cells.

Next, we tested the requirement for new T cell infiltration for the antitumor effects of a single 20-Gy dose delivered locally to MC38 tumors. These effects are dependent on CD8[+] T cell function[45]. The antitumor effects of 20 Gy were resistant to continuous treatment with FTY720 that started 3 h before IR (Fig. 6d). In contrast, FTY720 treatment beginning at the time of tumor inoculation abrogated the therapeutic effects of 20 Gy, confirming the need for T cell priming and subsequent development of effector function (Fig. 6d). This was further demonstrated by injecting a single dose of anti-CD8 antibodies after IR in mice treated with FTY720 to deplete all CD8[+] T cells including intratumoral T cells (Fig. 6e). Since FTY720 treatment prevents new T cell infiltration in these mice, the single dose of anti-CD8-depleting antibodies shows indirectly the contribution of intratumoral T cells to tumor control. The sufficiency of preexisting intratumoral T cells to control tumor growth following IR was confirmed in the Panc02SIYCerulean model (Supplementary Fig. 19). These results indicate that intratumoral T cells in irradiated tumors have a decreased proliferative potential, yet show an augmented ability to produce IFNγ and suffice to mediate the antitumoral effect of local SBRT doses, at least during the first few weeks after IR.

## Discussion
Here we show that the T cells present in inflamed/immunogenic solid tumors at the time of treatment are not eliminated by radiation doses and schedules typically used in the clinical setting, and T cells appear to be essential to the antitumor effects of

therapeutic IR. These results challenge the current paradigm that T cells are sensitive to killing or have their antitumor function decreased by IR. Activated and memory T cells are reported to be more radioresistant than naïve T cells[7–9]. These data could explain in part our observations with antigen-experienced tumor-infiltrating T cells. However, we detected a 1.5-fold cell loss for intratumoral T cells in Fig. 3d, as compared with the reported fivefold cell loss for activated/memory cells in spleen from mice at a similar dose of WBI[9]. This result suggests that the greater radio-resistance of intratumoral T cells cannot be fully explained by an antigen-experienced phenotype. The influence of the tumor (and certain non-lymphoid tissue) microenvironments could be key to fully understand this phenomenon. The critical role of $T_{RM}$ and even their quantitative contribution[46] to immunological memory were overlooked previously because memory responses had usually been measured in blood and lymphoid organs[47]. Similarly, the idea that IR would eliminate all T cells in the tumor derives from the exhaustive measurements of IR effects on the blood of cancer patients treated with IR, and in mouse lymphoid organs. Both blood and lymphoid organs contain the most sensitive T cell populations, as we show here. Therefore, the possible role of tissue and tumor-resident T cells in the response to IR has remained relatively under-investigated until now.

$T_{RM}$ and intratumoral T cells display intriguing similarities, including parenchymal localization (proven by intravascular staining exclusion, Fig. 3), and similar transcriptional profiles (Fig. 5 and see refs. [48,49]). From the tissues/organs studied here, tumors and IEL had the highest percentage of $T_{RM}$ and also the most radioresistant CD8[+] T cells; however, the resistance to IR was not limited to T cells with a $T_{RM}$ phenotype, but similar for T cells with or without $T_{RM}$ surface marker expression. In contrast, in the liver, a higher radio-resistance was associated with the $T_{RM}$ phenotype. This could be because a lower percentage of $T_{RM}$ results in a more diverse non-$T_{RM}$ T cell population in the liver, which includes radiosensitive T cells. Alternatively, the skin (where our s.c. tumors grew) and the gut mucosa could provide unique external pro-survival signals to all T cells within the tissue. Lastly, $T_{RM}$ and intratumoral T cells share a dependence on TGFβ

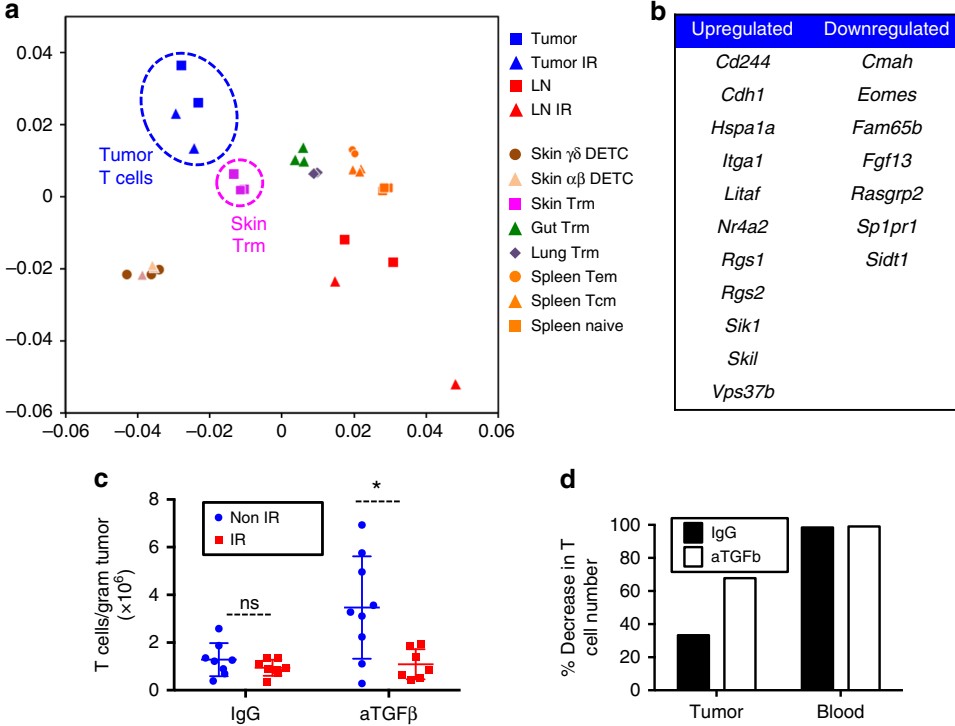

**Fig. 5** Intratumoral T cells are transcriptionally similar to $T_{RM}$ and are more sensitive to IR in mice treated with anti-TGFβ antibodies. Gene expression data from LN vs. tumor T cells (Fig. 4) was compared to published data on gene expression differences between $T_{RM}$ cells and spleen memory T cells (GEO GSE47045) to estimate similarity across all samples. **a** PCA-style plot shows that tumor T cells were phenotypically closest to skin and other $T_{RM}$, whereas LN T cells were closest to spleen naïve or memory T cells. **b** Eighteen of 37 genes from the published $T_{RM}$ signature were differentially expressed also in intratumoral T cells. **c** Total CD3+ T cells were quantified 24 h after 8 Gy WBI of mice bearing established MC38 tumors that had been treated or not with anti-TGFβ blocking antibodies every 2–3 days starting 2 days after tumor cell inoculation until day of sacrifice. Data are pooled from two independent experiments with a total of nine mice per group. The plot shows mean and SD. n.s., non-significant; *$P = 0.01$ (unpaired $t$-test). Each dot represents an individual mouse. **d** The effect of IR in mice from **c** was calculated as percent decrease in the average number of T cells per gram of tumor. The percent decrease in the average number of T cells per microliter of blood in a subset of mice from **c** that were bled before sacrifice ($N = 3$–4 per condition) is shown as a reference

for development and reprogramming, respectively. TGFβ signaling blockade leads to an increased number of T cells in unirradiated tumors at baseline, resulting in similar number of T cells/gram of tissue after IR when compared with control animals. However, the percentage of cells eliminated by IR is greater when TGFβ is blocked (Fig. 5d). This increase in T cell elimination might be a result of TGFβ blockade promoting T cell proliferation[50] and rendering T cells more sensitive to IR, but that does not seem likely since proliferative status of T cells did not predict radiosensitivity in our hands. Published evidence supports a radio-protective effect for TGFβ on other malignant and non-malignant cell types, through multiple mechanisms including an increased DNA damage response[51–53], prevention of a mitotic catastrophe, lower intrinsic free radical levels, or reduced activation of the death receptor pathways[54]. Our data suggest that TGFβ contributes to the radio-resistance of tumor-infiltrating T cells. This effect may need to be considered in clinical trials testing anti-TGFβ antibodies together with radiotherapy. More studies testing the length and timing of treatment with anti-TGFβ in animal tumor models will be necessary to determine optimum usage of anti-TGFβ with radiotherapy. Other factors that T cells in the tumor microenvironment are exposed to, such as hypoxia or integrin-mediated adhesion, are likely to also play a role in intratumoral T cell radio-resistance and will need to be examined separately.

Intratumoral T cells acquired patterns of gene expression commonly found in cancer cells. A possible explanation behind

the similarities of tumor-infiltrating T cells and $T_{RM}$ may be their ability to adapt to the microenvironment e.g., epidermal $T_{RM}$ acquire phenotypic characteristics similar to Langerhans cells and dendritic epidermal T cells residing in the same niche[32]. Therefore, intratumoral T cells might have different gene expression patterns and degrees of radio-resistance depending on tumor type and location. Interestingly, tumor-infiltrating T cells from MC38 colon carcinoma cells injected s.c. more closely resemble skin than gut $T_{RM}$ cells (Fig. 5). This raises the possibility that orthotopic colon cancer tumor-resident T cells would be transcriptionally closer to gut $T_{RM}$ cells, although recent studies show that tumor-infiltrating T cells from orthotopic mammary tumors and transplantable melanoma cluster together despite their different tissue origins[55].

Irradiated preexisting intratumoral T cells had a severely diminished proliferative capacity, as expected from radiation-induced DNA damage; however, they retained their motility and their ability to produce IFNγ, showing an increased production of IFNγ at 5–9 days after IR when compared with unirradiated cells. A myeloid cell infiltrate (Supplementary Fig. 17, 18) that is both less suppressive and more enriched in DCs expressing higher MHCII levels could contribute to explain such improved effector function, although other IR-induced changes in the microenvironment are likely to play a role as well. IFNγ production mediates T cell-induced tumor ischemia[56] and its increased production after IR could be key for tumor control by preexisting intratumoral T cells.

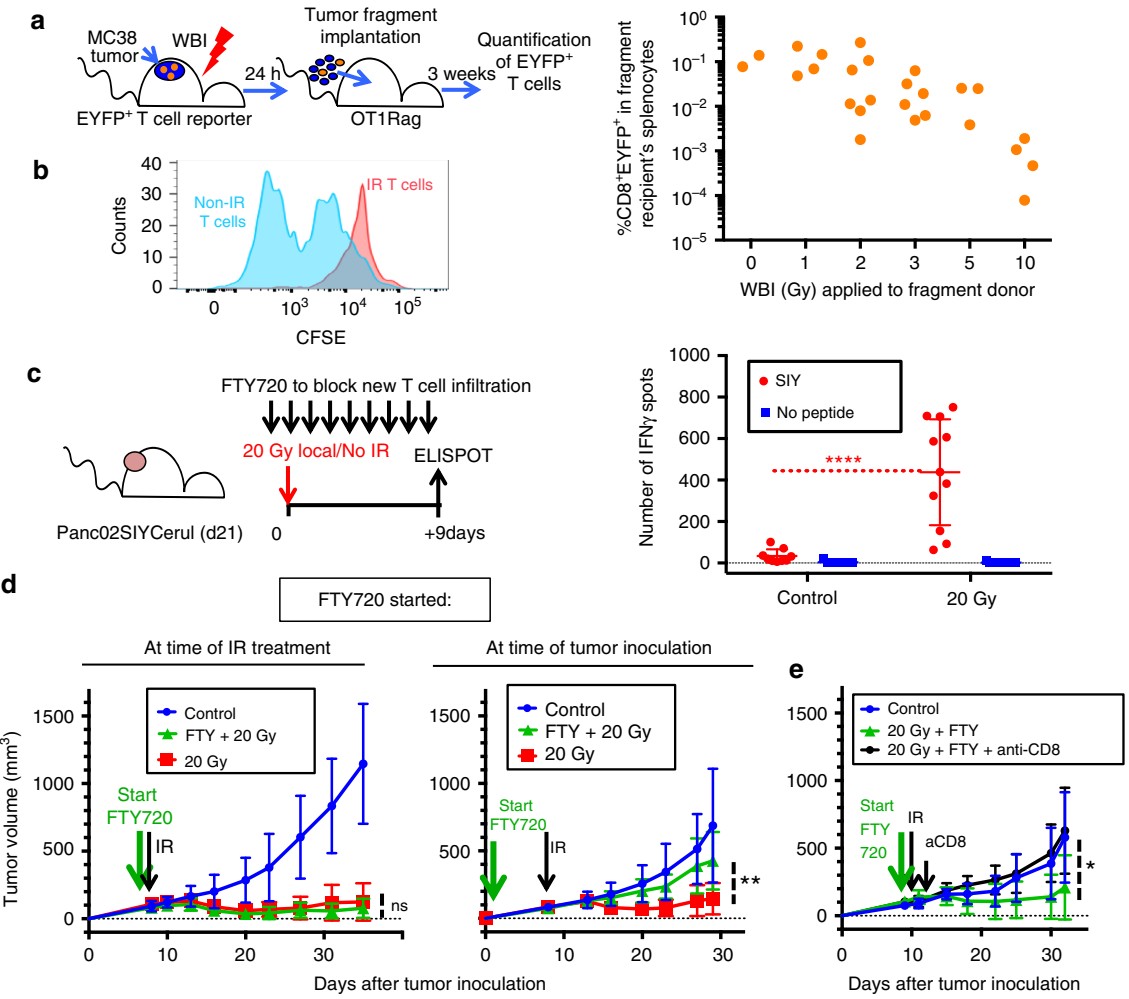

**Fig. 6** Irradiated intratumoral T cells produce IFNγ and mediate the therapeutic effects of 20 Gy in the absence of new T cell infiltration. **a** Tumor fragments from irradiated T cell reporter mice in Fig. 1 were transplanted into immunodeficient hosts 24 h after WBI. Three weeks later, the number of EYFP+ T cells that originated from the transplanted fragments was determined in the spleens of the recipient animals. Linear regression analysis showed a significant dose-dependent decrease ($Y = -0.01092*X + 0.09094$, $P = 0.0131$) **b** MC38 tumor-infiltrating CD8+ T cells were isolated from mice treated with 5 Gy WBI or non-irradiated, labeled with CFSE and stimulated with antiCD3/CD28 beads to measure their proliferation. Data are from T cells from three mice pooled per group and representative of two independent experiments. **c** IFNγ production by T cells purified from irradiated Panc02SIYCerulean tumors 9 days after 20 Gy local IR, by ELISPOT. Mice were treated with FTY720 every 24 h starting 24 h before IR to prevent new T cell infiltration. Data are pooled from three independent experiments with $N = 9$ (control) and $N = 11$ (20 Gy) total animals per group. ****$P < 0.0001$, Mann–Whitney test. **d** B6 mice bearing MC38 s.c. tumors were treated at days 8–9 after tumor cell inoculation with 20 Gy local IR. Some mice also received 20 μg FTY720 every 24 h, starting 3 h before IR or at the time of tumor inoculation, and until the last data point. **$P = 0.0075$. $N = 8$ mice per group. **e** Mice treated with FTY720 starting 3 h before IR, received a single dose of anti-CD8 depleting antibodies 24 h after IR. *$P = 0.05$ (**d**, **e**, unpaired t-test); data in **c** and **e** show average and SD

Our results show that preexisting tumor T cells survive IR and mediate antitumor responses. Peripheral newly tumor-infiltrating T cells likely also contribute to the antitumor effects and are important for at least two reasons: (i) irradiated preexisting T cells show a compromised proliferative capacity, which probably affects their ability to become memory cells. Newly infiltrating T cells can be reprogrammed as tumor-resident T cells with an intact memory capacity. (ii) The tumor models used here (MC38 and Panc02SIY EGFP) have some baseline T cell infiltration, similar to human "inflamed" tumors. For tumors that have no infiltrating T cells ("non-inflamed"), irradiation could be a way to attract newly infiltrating peripheral T cells to make them more "inflamed" and able to respond to immunotherapy. Consistent with this notion, in the context of checkpoint blockade/IR combinations, resident T cells contributed significantly to the therapeutic effect[57,58], although newly infiltrating T cells seemed to be required for optimal effects, especially when using IR and PD1 blockade[58].

Overall, our findings indicate that local tumor T cell status contributes to the success or failure of radiotherapy. Our own recent findings from clinical trials suggest that anti-PD1 therapy significantly enhances the local effect of SBRT[59]. Our present study reveals one potential mechanism: the activation of local immune responses in the irradiated tumor. These data also support our ongoing clinical trials testing whether treating all or many metastases[60] with radiotherapy plus/minus checkpoint inhibitors may improve clinical outcomes. These clinical trials are in large part based on the findings reported herein as well as the concept that the overall tumor cell-to-T cell burden might be determinative[61] in a robust systemic response.

## Methods

**Mice and in vivo treatments.** T cell reporter mice were generated by crossing Lck-cre (JAX 003802) or CD4-cre (JAX 017336) with R26R-EYFP (JAX 006148) mice, all from Jackson. Six- to 8-week-old C57BL/6 female mice were purchased from Jackson or Harlan. OT1$Rag^{-/-}$ and 2C-Thy1.1 mice were kindly donated by Dr. Hans Schreiber (University of Chicago). $1 \times 10^6$ MC38 tumor cells or $2.5-5 \times 10^6$ Panc02SIYCerulean cells were subcutaneously injected into the flank of mice. In experiments using Panc02SIYCerulean, we injected a single 200 μg dose of anti-CD8 antibodies (10F.9G2; BioXCell) on day 0, i.p. For tumor fragment transplantation, established MC38 tumors grown in T cell reporter mice were excised, diced into 1–2 mm fragments into RPMI, and implanted subcutaneously into naive C57BL/6 mice using a 13-gauge trochar (between 50 and 200 μL fragments suspension/recipient). Tumor volumes were measured along three orthogonal axes (a, b, and c) and calculated as tumor volume = abc/2. FTY720 (Enzo) or vehicle control (DMSO) were administered by oral gavage at 20 μg every 24 h. FTY720 stock solution aliquots (10 mg/mL in DMSO) were kept at −20C and diluted to a final concentration of 125 μg/mL in phosphate-buffered saline (PBS) directly before administration. In vivo TGFβ-blocking antibody (1D11.16.8, Cat. No. BE0057) and mouse IgG1 isotype control antibodies (BE 0083) were from BioXCell and were administered at 100 μg/dose, i.p. Mice used in longitudinal imaging experiments were injected i.v. with $5 \times 10^6$ bone marrow cells from DsRed$^+$Rag$^{-/-}$ mice within 24 h after WBI. Splenocytes from 2C-EGFP or 2C-Thy1.1 transgenic mice were stimulated in vitro with 1 μM of SIYRYYGL (SIY) peptide at $4 \times 10^6$ cells/mL and 3 mL/well in six- well plates. After 3–4 days, each recipient mouse received the pre-activated T cells from one well (between 10 and $18 \times 10^6$ cells) i.v. into the retroorbital plexus. In one experiment for Fig. 1, 2C T cells expressed EGFP by retroviral transduction with the pMP71-EGFP plasmid[62]; T cells from 2C-EGFP transgenic mice were used otherwise. All mice were maintained under specific pathogen-free conditions. Animal care and use were in accordance with institutional and NIH protocols and guidelines. All studies were approved by the Animal Care and Use Committee of The University of Chicago.

**Functional assessment of T cell and myeloid cell function.** For assessment of tumor-infiltrating T cell proliferation, CD8$^+$ T cells were magnetically purified from MC38 tumors using Miltenyi mouse CD8$^+$ positive selection kit, stained with CFSE (SIGMA) 5 μM for 15 min at 37 °C and stimulated for 3 days with Dynabeads Mouse T-Activator CD3/CD28 beads, according to the manufacturer's instructions. For IFNγ ELISPOT, 96-well HTS-IP plates (Millipore) were precoated with capture anti-IFN-γ antibody (clone R4-6A2) overnight at 4 °C. For T cell function experiments, $0.4–2 \times 10^5$ CD8$^+$ T cells purified (Miltenyi) from Panc02SIY Cerulean tumors and irradiated (12 Gy) B6 splenocytes were added at a 1:1 ratio, with or without SIY peptide (1 μM). To test tumor APC function, we isolated CD11c$^+$ cells (EasySep Mouse CD11c Positive Selection Kit II; Stemcell Technologies) from the flow through resulting after CD8$^+$ T cell purification from tumor cell suspensions. We compared different ratios of naïve 2C:APC cells, keeping constant the number of 2C cells ($2 \times 10^5$). After 72 h incubation at 37C, cells were removed and the ELISPOT plates were incubated for 2 h at 37 °C with biotinylated anti-IFN-γ antibody (clone XMG1.2), and subsequently with avidin–horseradish peroxidase for 1 h at room temperature. Spots were developed by the addition of AEC substrate (all ELISPOT reagents from BD Pharmingen) and quantitated using a CTL-ImmunoSpot S6 Core Analyzer from Cellular Technology Ltd (Cleveland, OH). To test intratumoral myeloid cell suppressive function, we isolated CD11b$^+$ cells (Miltenyi) from tumors at different times after IR and incubated them with OT1 splenocytes labeled with CFSE and stimulated in vitro with OVA peptide (SIINFEKL) 1 μM for 4 days. The ratio of CD11b$^+$ cell: OT1 splenocyte was 1:1. Suppression of OT1 proliferation was calculated over control wells that had no CD11b$^+$ cells added, considering OT1 cells that had undergone at least two cycles of proliferation.

**Cell lines and reagents.** Panc02 murine pancreatic adenocarcinoma cells donated by Michael Gough (Oregon Health and Science University) were retrovirally infected with pMFG (SIY)$_3$-Cerulean kindly provided by Dr. Hans Schreiber[34]. Briefly, Phoenix-ampho cells were transfected with the retroviral vector plasmid using the CalPhos Mammalian Transfection Kit (Clontech, Mountain View, CA), and supernatants were then used to transduce Panc02 cells. After infection, cells were FACS-sorted for high expression of SIY-Cerulean to generate the Panc02-SIYCerulean cell line. The MC38 colon adenocarcinoma cell line was kindly provided by Dr. Yang-Xin Fu. MC38-Cerulean cells[33] were kindly donated by Dr. Hans Schreiber. Cells were used after 1–4 weeks in culture.

**Mouse irradiation.** Mice were irradiated using a Phillips IR250 orthovoltage X-ray generator operating at 250 kV 15 mA. For tumor irradiation, each mouse was confined to a cylindrical lead cover with its tumor-bearing flank exposed through an opening on the side, allowing the tumor to be irradiated locally. For WBI excluding the tumor, the lead cover was used to shield exclusively the tumor-containing window chamber, while the rest of the body was exposed to IR. For some experiments involving myelo- ablative WBI, a dose of 8 Gy was used because it achieves quick and durable lymphocyte depletion in blood[63] while causing minimum radiation-induced toxicity[64].

**Longitudinal in vivo imaging of tumors.** Dorsal skinfold windows were surgically implanted into the backs of anesthetized T cell reporter (Lck-EYFP in most experiments) mice, as in ref. [34]. Immediately after surgical implantation of the window, $1–5 \times 10^6$ Cerulean-expressing tumor cells were placed within it. A glass window was placed to cover the exposed tissue and secured with a snap C-ring. Tumor development was monitored by fluorescent confocal microscopy using a Leica SP5 II TCS tandem scanner 2-photon spectral confocal microscope with XY motorized stage. Long-working distance ×20/NA 0.45 and ×4/NA 0.16 dry lenses were from Olympus. Excitation wavelengths were: CFP 458, EGFP 488, EYFP 514, DsRed 561. Narrow emission windows were used at peak emission to minimize cross-talk of probes, e.g. [458,463–484; 476,496–505; 514,519–544; 561,574–654] [ex,em]. Mice were anesthetized with inhaled isofluorane during the imaging sessions. During the first imaging session, within 1 week after surgery, all areas (usually 2–5) containing live fluorescent cancer cells within the 1 cm diameter window were imaged at low-magnification (×4). Within those ×4 regions, about 2–6 representative ×20 images per day and mouse were taken. ×20 regions were selected on the basis of (i) showing vasculature, (ii) being as distant as possible from each other, to thoroughly cover the window. The same initial ×4 and ×20 areas were revisited and imaged as long as viable during the subsequent sessions. If the original ×4 or ×20 regions became non-viable at some point during the experiment, and any new live regions had developed in the meantime, these new regions were subsequently tracked instead. Data from only identical regions longitudinally tracked before/after IR (within a $1.2 \times 0.8$ mm tissue region) were plotted in Fig. 2. All preexisting (EYFP$^+$) and newly infiltrating T cells (EGFP$^+$) within those regions were tracked. Numbers of imaged T cells used to create the quantitative graphs in Fig. 2 varied between 114 and 471 (EYFP$^+$) and 10 and 104 (EGFP$^+$) per mouse and time point. The tumors in our imaging system grow unimpeded, and when they reach large size the center of the tumor becomes necrotic, which prevented us from continuing imaging beyond $34 \pm 4$ days in untreated or $40 \pm 1$ days in irradiated mice, when viable tissue was no longer found within the window. The depth of penetration into the tissue that is reached for imaging with this technique is approximately 100–300 μm.

**Quantitative analysis of images.** Initial digital image processing was performed using Leica LAS-AF Lite and selected images were further analyzed using Fiji (NIH). A macro was created in Fiji for the automated quantification of EYFP$^+$ and EGFP$^+$ cells in cross-talk corrected ×20 images taken before/after IR. EYFP$^+$ and EGFP$^+$ T cell counts were then normalized relative to the maximum count observed in each longitudinal experiment, and this value was considered 100%. Maximum counts in the experiments shown in Fig. 1 corresponded to 346 EGFP$^+$ and 339 EYFP$^+$ T cells (SBRT model, using transgenic 2C-EGFP$^+$ T cells) and 106 EGFP$^+$/325 EYFP$^+$ T cells (fractionated IR model, using bulk retrovirally-transduced EGFP$^+$ 2C T cells). Imaris 8.4.1 software (Bitplane) was used for T cell motility analysis and quantification. Approximately 20 min-duration xyzt stacks were corrected for drift based on landmark features and for cross-talk between channels. Arrest coefficient was calculated as the fraction of time that T cell velocity was less than 3 μm/min.

**Tissue processing, flow cytometry, and cell sorting.** Tumors were excised and digested for 20–45 min at 37 °C with 75 μg/mL liberase TM (Roche) and 20 μg/mL DNase I (Sigma), pipetted up and down for about 2 min with a plastic 3 mL pipette and filtered through a 70-μm nylon mesh filter to generate single-cell suspensions. In some experiments, this was followed by magnetic enrichment of certain populations using microbeads specific for CD8$^+$, CD45$^+$, CD11b$^+$ (Miltenyi, positive selection kits), or CD11c$^+$ cells (EasySep; StemCell), as indicated in other parts of the manuscript. Lymph nodes draining and not draining the region of implanted tumors had to be pooled to obtain sufficient cells for experiments. For sorting experiments, inguinal and axillary lymph nodes were pooled; for tissue-resident T cell determination experiments, only inguinal lymph nodes were used. In all cases, mice had been implanted with a tumor on the right flank. For studies on tissue-resident T cells, parenchymal (non-circulating) CD8$^+$ T cells were distinguished from circulating T cells trapped in the organ vasculature using the intravascular staining technique[37]. Briefly, mice were injected with 3 μg of anti-CD8α-APC antibody i.v. 3–5 min before sacrifice. Organs were isolated after perfusion with PBS with 75 U/mL heparin. Spleens, lymph nodes and livers were mechanically disrupted through a sterile 70-μm nylon mesh filter. Hepatocytes were excluded from the liver samples by performing a single-step 35% Percoll centrifugation (GE Healthcare, Chalfont St Giles, UK) and using only the pelleted cells. Lungs were diced into 1 mm pieces, digested with 125 μg/mL liberase TM and 20 μg/mL DNase I for 20 min at 37 °C, passed through a sterile 70-μm nylon mesh filter and spun down in 40% Percoll (adapted from ref. [65]). Erythrocytes from spleens, lungs, and livers were lysed with ACK buffer. For extraction of IEL, small intestines were cut into 1 cm long pieces after removal of Peyer's patches, and fragments were incubated for 20 min at 37 °C under gentle shaking in complete RPMI media (10% fetal bovine serum, HEPES, non-essential aminoacids, ʟ-glutamine, 2-mercaptoethanol) with 5 mM EDTA and 0.145 mg/mL DTT. Fragments of intestine were then filtered using a strainer and further stripped of the epithelium containing the IELs through subsequent rounds of vigorous shaking in a 50 mL conical tube with wash buffer (RPMI supplemented with HEPES and penicillin/streptomycin), followed by filtering and centrifugation of the flow

through. Single-cell suspensions from all organs were stained with relevant antibodies for 15 min at 4 °C, and washed twice with cold PBS. Blood samples were stained, treated with $NH_4Cl$ red blood cell lysis buffer, and immediately acquired (without washing) after the addition of BrightCount beads (Invitrogen) to determine the absolute counts of cell populations in PBL. For sorting of $CD8^+$ $CD44^+CD62L^-$ T cells from tumors, cell suspensions obtained by enzymatic digestion were spun down in 35% Percoll and enriched for $CD8^+$ T cells using EasySep $CD8^+$ positive selection kit before staining with appropriate antibodies and FACS-sorting.

Antibodies for mouse CD8α (clone 53-6.7), CD8β (YTS156.7.7), CD4 (RM4–5), CD3 (17A2), CD44 (IM7), CD62L (MEL-14), α1 (HMα1), αV(RMV-7), αM /CD11b (M1/70), F4/80 (BM8), Ly6C (HK 1.4), Ly6G (1A8), CD103 (2EX), CD69 (H1.2F3), Ki67 (SolA15), CD39 (24DMS1), PD1 (29F.1A12), phospho-histone H2AX (CR55T33), LFA1 (H155–78), FoxP3 (FJK-16s), LAG3 (C9B7W), TIGIT (GIGD7), 41BB (17B5), NK1.1 (PK136), B220 (RA3-6B2), F4/80 (BM8), CD11c (N418), I-A/I-E (M5/114.15.2), Thy1.1 (OX-7), Thy1.2 (53–2.1) were from BioLegend or eBioscience. Ki67 and phospho-histone H2AX were stained using the eBioscience transcription factor staining buffer set according to the manufacturer's instructions. Data were acquired on a LSRII or LSR-Fortessa (BD) and analyzed with FlowJo software. Sorting was performed on a FACSAriaII (BD) at the Flow Cytometry Facility of The University of Chicago. Purity of sorted fractions is typically around 99.5%.

**Gene expression analysis**. RNA was extracted from sorted $CD8^+CD44^+CD62L^-$ T cells from tumors and LN using RNA-Easy Plus Micro kit (Qiagen), according to the manufacturer's instructions. Gene expression analysis was performed using Affymetrix Mouse 430 2.0 GeneChips® at The University of Chicago Genomics Facility. Data were normalized using RMA approach[66], and differentially expressed genes between tumor-associated and LN-associated T cells or between non-irradiated and irradiated T cells were detected using two different approaches. First we used R/Bioconductor package limma (https://bioconductor.org/packages/release/bioc/html/limma.html) by fitting a robust linear model to normalized intensities values for all available probesets annotated to mouse genome. Probes were subsequently ranked for differential expression using the empirical Bayes method. Multiple testing corrections were performed using the Benjamini–Hochberg procedure. Differentially expressed mRNA were identified using an adjusted $P$ value of less than 0.1 and an absolute log2 fold-change equal to or greater than 1. In the second approach we used Significance Analysis of Microarrays (SAM)[67] for Excel (Stanford University, CA) with a False Discovery Rate (FDR) of 1% and a fold-change threshold of ≥2.0 (refs. [68,69]). Overlap of DEGs identified by the two independent approaches was used for further downstream analysis. To explore the functional significance of identified sets of DEGs in terms of biological processes, a Gene Ontology (GO) overrepresentation analysis was performed using online tools and databases provided by the Gene Ontology Consortium (db-Biological Processes, rel. 2017-04-24) (http://geneontology.org/) and the PANTHER Classification System (Overrepresentation Test, rel. 2017-04-13) (http://pantherdb.org/). Enriched annotations with significant $P$ values corrected by the Bonferroni method were retained and rank ordered by descending log2 fold-change for subsequent interpretation and visualization. Gene set expression analysis was performed using EGSEA v1.6.1 using probe annotations from mouse4302cdf v2.18.0 and mouse4302.db v.3.2.3 mapped to Broad Institutes molecular signatures database (MSigDB) v6.2 pathways.

**Gene expression similarities between T$_{RM}$ and tumor T cells**. To compare expression patterns of T$_{RM}$ cells with tumor-infiltrating T cells we used GEO GSE47045 dataset based on Affymetrix Mouse ST 1.0 and our data reported here based on Affymetrix Mouse 430 2.0 platforms. The raw probeset intensities were background subtracted, log2 transformed, and quantile normalized to adjust for variation that arise from microarray hybridizations for both datasets separately via RMA algorithm[66] from bioconductor package oligo[70]. To compare the expression profiles of different datasets from different profiling platforms, the platform batch effects were normalized with Bioconductor package SVA[71] via ComBat[72]. Biological annotations were obtained with oligo wrapper function getNetAffx. The expression profiles from different sequencing platforms were integrated together at the same transcript level. After the batch effect correction, the transcript expression profiles from two different platforms became balanced for further comparison analysis. The principal component analysis was then applied to visualize the grouping of samples.

**Statistical analysis**. Statistical analysis of T cell motility, %CD8$^+$ T cells in tumor and blood after WBI, suppression of T cell proliferation, changes in myeloid cell composition or surface/intracellular marker expression (unpaired $t$-tests, two-tailed), FTY720 blockade or TGFβ blockade in vivo (unpaired $t$-test with Welch's correction at last measurement time point), integrin expression in tumor vs. LN (paired $t$-test), effect of IR in parenchymal CD8$^+$ T cells from different organs, and in T$_{RM}$ vs. Non-T$_{RM}$ CD8$^+$ T cells (ratio-paired $t$-test), and linear regression were performed in GraphPad Prism 6.0. for Mac, GraphPad Software, La Jolla, California, USA. All $t$-tests were two-tailed. *$P \leq$ 0.05; **$P \leq$ 0.01; ***$P \leq$ 0.001; ****$P \leq$ 0.0001. For the analysis of T cell counts per

FOV (Fig. 1), data were plotted with a superimposed smooth curve estimate for the trend for each treatment group. Since a quadratic model was seen as an appropriate simple model for the response, a quadratic regression model was fit to estimate the trend function for each treatment group. A linear model was also fit for each response (yellow count and green count), with time since IR as a factor nested within the treatment groups. Either approach showed that the average of yellow and green counts over time were positive with a 95% confidence level. These analyses were performed using the statistical software R (version 3.2.2, https://www.r-project.org).

**Reporting Summary**. Further information on research design is available in the Nature Research Reporting Summary linked to this article.

## Data availability

Microarray data generated in this study have been deposited in Gene Expression Omnibus (GEO) with accession number GSE111492. All other data that support the findings of this study are available from the corresponding author on reasonable request.

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

## Acknowledgements

We thank Rolando Torres and Ani Solanki for technical assistance, as well as Jorge Andrade and Li Yan (Bioinformatics Core), Mihai Giurcanu (Biostatistics Lab), Christine Labno and Vytas Bindokas (Integrated Light Microscopy Core), Pieter Faber (Genomics Facility), David Leclerc (Cytometry and Antibody Technology) at the University of Chicago. We also thank Dr. Hans Schreiber for the donation of materials and mice; Carlos Martinez for help with the bioinformatics analysis; Laura Mackay, Peter Savage, Richard Hynes, Brendan MacNabb, Hua Liang, Meng Xu, and Samuel Hellman for helpful discussions, and Amy K. Huser for editorial help. This work was supported by funds from the Ludwig Foundation for Cancer Research and from Regeneron Pharmaceuticals.

## Author contributions

A.A. designed and conducted most of the experiments and data analysis and prepared the manuscript; C.F., M.B., W.Z. and B.B. helped with organ and cell isolation and tumor size monitoring; Y.H. and H.M. assisted with mouse work and live imaging experiments, S.J.C., J.J.L., I.L. and T.S. provided helpful suggestions, discussed data interpretation and contributed to the manuscript; S.P., M.F. and N.K. performed gene expression analysis; Y.-X.F. designed experiments and interpreted data; and R.R.W. supervised this work and helped to write the manuscript.

## Additional information

**Competing interests:** The authors declare no competing interests.

