## [Peer Review File · Nature Communications]

Reviewers' comments:

Reviewer #1, expert in anti-tumor T cell responses (Remarks to the Author):

Using a number of different models and approaches, Weichselbaum and colleagues have investigated the role of tumour-residing T cells in radiotherapy. They provide evidence that these T cells, as well as tissue-resident memory T cells in other peripheral locations, display a level of resistance to the depleting effect of irradiation (IR), delivered either locally or as whole-body irradiation. Transcriptional and functional analyses identify TGF-beta as an upstream regulator of irradiation resistance. Furthermore, they show that the function of tumour-residing T cells is preserved after IR and that these cells play an important role in tumour control after local therapeutic IR. In summary, the authors provide a number of important and compelling findings with critical relevance to clinical radio-oncology. I only have a few queries/suggestions.

Fig 1C+D. What exactly is shown as "Percent maximum number of T cells per FOV" on the y-axis. What is 100% - does this correspond to all (both resident and infiltrated) T cells per field? Related to this, the authors state that a "substantial fraction of tumour-resident T cells" were preserved after IR. Has this fraction been quantified relative to pre-IR numbers?

Given that TGF-beta is a major upstream regulator, it may be somewhat surprising that the tumour-resident T cells do not express CD103, a molecule induced by this cytokine in activated T cells. The authors should consider to mention and discuss this finding.

Lines 150-151: The authors state that "lymph node T cells... are enriched in non-recirculating CD8+ T cells (...average 97.2%)". How were these identified as non-recirculating? Is this based on CD69 expression? Also, do the lymph nodes drain the region of implanted tumours?

Fig 6A: What does "%CD8+EYFP+ detected in fragment recipient" on the y-axis refer to? Is this the percentage of whole splenocytes?

Line 122: "CD11c+ T cells". Typo, should be non-T cells, i.e. APC?

Does anti-CD8 Ab treatment ablate tumour-resident T cells – has this been quantified/shown? Alternatively, or in addition, it may act to block CD8 T cells function.

Reviewer #2, expert in cancer immunotherapy and radiation therapy (Remarks to the Author):

This manuscript addresses a very important issue, namely the survival of T cells present within a tumor treated with radiotherapy. Whereas it is known that T cells exhibit different radiosensitivity depending on their maturation and differentiation/activation status, whole body irradiation at relatively low doses has been used to eliminate T cells in clinical scenarios where it is necessary to "create space", such as in the setting of adoptive T cell therapy. Thus, the commonly held notion is that focal radiation used at therapeutic doses will kill most of the T cells infiltrating a tumor. In the new era of cancer immunotherapy, this has become especially important and a factor influencing the design of clinical studies of radiotherapy and immunotherapy combinations.

To address the fate of T cells that are present within the tumor microenvironment at the time of radiotherapy treatment, Arina and colleagues have developed an elegant model based on the T-cell reporter (Lck-EYFP) mouse that enabled them to distinguish between pre-existing and newly primed/recruited T cells following radiotherapy. They show that many T cells survive even after a 20Gy radiation dose, and retain motility and effector functions while losing the ability to proliferate.

To explain the differential sensitivity to radiation of tumor-residing T cells, circulating T cells, and T

cells residing in lymph nodes and spleen, a number of experiments are performed. The main conclusion that tumor-residing T cells are similar to tissue-resident memory (TRM) T cells is consistent with a growing literature about the presence and role of TRMs in tumors, in both mouse models and patients. However, this conclusion is based on gene expression analysis of bulk sorted CD8 T cells and does not allow the assessment of what percentage of the intratumoral T cells are TRMs. The heterogeneity of the tumor-residing T cells is only superficially investigated, as is the potential influence of other immune cells on T cell survival and functionality in irradiated tumors. Mechanistically, the claim that TGF β is responsible for determining T cell radiosensitivity is not fully supported by the data, as detailed in specific comments below. As mentioned in discussion, TGF β is the target of some immune-oncology strategies, due to its immunosuppressive role. New bifunctional agents (e.g. TRAP, targeting PDL-1 and TGF β) are undergoing clinical testing. Thus, it is important to determine if this pleiotropic factor plays a role in T cell radiosensitivity, but also critical to provide solid evidence as the stakes are high for the development and optimal use of TGF β -targeting therapies. Finally, tumor-residing T cells are likely to be an important mechanism influencing the response of the tumor to SBRT only in the case of "hot" tumors. For poorly immunogenic mouse tumors, and for the majority of cancer patients, it will be far more important if radiation can generate new T cells.

Overall, the data provided about the sensitivity of T cells present within tumors to radiation are relatively convincing and have the potential to advance the field. However, there are several weaknesses on the mechanistic aspects of the study that need to be addressed.

Major issues:

- 1) On page 4 the statement: "A single 200 μ g dose of anti-CD8 depleting antibodies on day 0 allowed the tumors to grow unimpeded by any "vaccination effects" induced by SIY-expressing tumor cell inoculation". This conclusion is not justified. Supplementary Fig 1 shows that in CD8-depleted mice the tumors grow faster and in all mice. However, the large number of T cells present in these tumors in untreated mice, shown in Fig 1 indicates that during tumor growth anti-tumor immunity develops. Presumably CD8 T cell numbers recover quickly after the initial depletion. The immune infiltrate should be characterized at the time of irradiation for relative presence of CD4 and CD8 T cells, Tregs, and their activation/exhaustion state. Some characterization is provided in Supplementary Fig 3, but this is limited to a few markers and to CD8 T cells. Markers of resident memory T cells are not included.
- 2) Supplementary Fig 3: Why only PD1 and CD39 were analyzed to assess the effects of irradiation on the endogenous and adoptively transferred CD8 T cells? The similar percentage of H2AX γ foci in the endogenous and adoptively transferred CD8 T cells suggests that most of the adoptively transferred CD8 T cells were already in the tumor at the time of radiation. The small difference in CD39 and PD1 expression may simply reflect the fact that the adoptively transferred CD8 T cells all bear the same TCR and recognize the same antigen, whereas the endogenous T cells are a heterogeneous population.
- 3) Supplementary Fig 4A: Why such strange radiation protocol of 10Gy + 20Gy 4 days later is chosen for another tumor model, MC38? Is this clinically relevant? Since the data are based only on the detection of EYFP+ cells by imaging, can it be ruled out that some of the positivity seen after radiation reflects phagocytosis of dying T cells by myeloid cells rather than surviving T cells? Without more evidence the statement that "at all time points, including those obtained after the 20 Gy dose, tumor resident T cells were detectable" is not fully warranted.
- 4) Figure 3A: this figure makes an important point about the radiation resistance of T cells present in the tumor versus the blood. However, data are presented as "% T cells" and it is unclear what the denominator is. The number and not the percentage of T cells should be shown.
- 5) Figure 3B and C: CD4 T cells should be further separated into regulatory and non-Treg, given data that they have a differential radiosensitivity (Kachikwu, E. L., Int J Radiat Oncol Biol Phys 2011).

6) Figure 3D: None of the comparisons between controls and irradiated mice seems significant due to the large variability in controls. Some correction for organ size/weight may reduce the variability. Moreover, the lungs contain several peri-bronchial lymph nodes and bronchial-associated lymphoid tissue, which, like spleen and LN will contain a large fraction of naïve T cells. The latter are known to be more radio-sensitive than memory T cells. In fact, in the lungs there is a marked reduction in CD8 T cells after IR. The variable inclusion of lymph nodes/lymphoid tissue in each of the organs may explain the big variability seen even in the absence of treatment. Overall, this analysis is too superficial to support the conclusion that the "organ microenvironment" determines T cell radiosensitivity. Despite the high CD69 expression, there are very few CD103+ cells in the tumor (Supplementary Fig 4B). Thus, another variable is the type of T cells that are present in each organ/tissue site analyzed (e.g., TRM versus T effector memory, etc...). Additional markers, including markers known to define naïve and TRM in different organs, (e.g., LFA1 in liver) would help support a more cautious interpretation of the results.

7) Page 7: it is not clear what are the data supporting the statement that "Like tumor T cells, lymph node T cells are enriched in non-circulating CD8+ cells" in Figure 3D?

8) Supplementary Fig 5: The analysis of the molecular pathways activated by IR should include statistical significance.

9) Supplementary Figure 6 lacks a legend to indicate the magnitude of change in expression of the genes in the heat map.

10) Supplementary Fig 7: The analysis of the molecular pathways activated in tumor vs lymph nodes should include statistical significance analyses. The legend lacks sufficient information to understand what is shown and how the analyses were performed.

11) On Page 8, the statement "Functional analysis detected TGF β as the top upstream regulator of T cell reprogramming in the tumor microenvironment" is only supported by z score and p value. It is not clear how this conclusion is supported by data. For instance, metabolic re-programming of T cells in the TME has been reported in several publications, and in Suppl Table 1 Hypoxia ranks a lot higher than TGF β signaling. Moreover, TGF- β is known to drive CD103 expression in TRM T cells (Nizad et al., Nat Comm 2017) but CD103 was barely expressed by the intratumoral CD8 T cells studied here (Supplementary Fig 4B). In addition to addressing these issues, the pathways that are under control of TGF β and are selectively activated in the tumor T cells should be shown.

12) Fig 5A shows that the tumor CD8 T cells are more similar to TRM T cells than to spleen T cells, something that is not surprising, and also not very novel.

13) Data in Fig. 5C do not demonstrate that blocking TGF β increases radiation sensitivity of the T cells present in the tumor. TGF β was blocked starting 2 days after MC38 inoculation. This led to a large increase in T cells density within the tumor (did it also reduce tumor growth?). Thus, at the time when radiation is used, the tumors in mice treated with TGF β blockade may be different in many other ways from the untreated tumors, and without data to show that the populations of T cells present are phenotypically and functionally similar the conclusion that TGF β controls radiosensitivity is not warranted. In Supplementary Fig 9, the differences do not appear to be statistically significant. Experiments addressing in more depth the role of TGF β signaling on T cell response to radiation are needed.

14) Supplementary Figure 12B: there is no statistically significant change in ability of CD11c+ cells to stimulate naïve T cells at day 5 post-irradiation. The text should be corrected to recognize this, or additional experiments performed to determine if the CD11c+ DC are more stimulatory. Importantly, it should be shown if DC numbers and phenotype are altered by radiation.

15) Supplementary Figure 12 E: Changes in MDSC are compared to non-IR tumors. However, over the course of 9 days, much can change also in the absence of treatment as tumors progress. Thus, IR and non-IR tumors should be compared for each of the days shown.

16) Discussion:

In the sentence "Here we show that the T cells present in solid tumors at the time of treatment, are not eliminated by radiation doses and schedules typically used in the clinical setting, and appear to be essential to the anti tumor effects of therapeutic IR." The second part is true only if there are sufficient T cells in the tumor at the time of IR. In many cases in patients this is not the case.

Minor issues:

- 1) Page 9, line 220: remove "T" after CD11c+
- 2) Fig 1 legend for (D): the explanation of the "N of optical regions" is unclear.
- 3) Figure 2 C and E: the different colors used for T cell tracks should be defined as to their meaning
- 4) Supplementary Fig 3: the flow plots in D are not informative as shown. Gating strategy should be shown.
- 5) Supplementary Fig 3 B and C: define what red and blue colors indicate

Response to reviewers NCOMMS-18-37757, Arina et al. "Tumor-reprogrammed resident T cells resist radiation to control tumors"

We thank the Editor and the Reviewers for their comments, which have enabled us to improve the quality of the manuscript. We believe that we have satisfactorily addressed the questions from both reviewers and hope that the paper can be now considered for publication in *Nature Communications*. The Reviewers' comments appear in blue and italics. Changes in the manuscript have been highlighted to facilitate review.

Reviewer #1

Using a number of different models and approaches, Weichselbaum and colleagues have investigated the role of tumour-residing T cells in radiotherapy. They provide evidence that these T cells, as well as tissue-resident memory T cells in other peripheral locations, display a level of resistance to the depleting effect of irradiation (IR), delivered either locally or as whole-body irradiation. Transcriptional and functional analyses identify TGF-beta as an upstream regulator of irradiation resistance. Furthermore, they show that the function of tumour-residing T cells is preserved after IR and that these cells play an important role in tumour control after local therapeutic IR. In summary, the authors provide a number of important and compelling findings with critical relevance to clinical radio-oncology. I only have a few queries/suggestions.

Fig 1C+D. What exactly is shown as "Percent maximum number of T cells per FOV" on the y-axis. What is 100% - does this correspond to all (both resident and infiltrated) T cells per field?

T cell counts, for resident and newly-infiltrated T cells separately, were normalized to the highest value observed in each experiment. For example, in the longitudinal imaging experiment using an SBRT model (Fig 1C, bottom), the highest count of EGFP cells observed was 346 cells per field of view (FOV) at day 14 since adoptive transfer in 1 out of 10 total regions measured at that time point for the IR mouse. Therefore, the maximum cell count (i.e., 346) was made to equal 100% and all other values for EGFP counts in that experiment were normalized to that value.

Normalization was used to (i) emphasize the changes induced by IR in pre-existing and infiltrating T-cell levels, and (ii) avoid confusion between transduced vs. transgenic expression of EGFP by 2C cells in the different experiments: i.e., since transduction was not 100% efficient and bulk EGFP-transduced 2C cells were transferred in the fractionated IR experiment shown in Fig 1C (top), lower absolute EGFP⁺ cell counts compared with the SBRT experiment using transgenically labeled 2C-EGFP cells (bottom) could be erroneously interpreted as different degree of infiltration due to the different IR regimes. After internal presentations among members of our team, it was decided that normalization would be best to avoid confusion. Normalization procedure and maximum counts observed for each experiment and each channel are now detailed in the Materials and Methods section (p.19).

Related to this, the authors state that a “substantial fraction of tumour-resident T cells” were preserved after IR. Has this fraction been quantified relative to pre-IR numbers?

Yes, we have quantified this fraction and now provide the numbers in the main text (p.5). The average resident T cell numbers detected in the last time point measured were 85% and 65% of the initial pre-IR EYFP⁺ T cell counts, for fractionated and SBRT models, respectively.

Given that TGF-beta is a major upstream regulator, it may be somewhat surprising that the tumour-resident T cells do not express CD103, a molecule induced by this cytokine in activated T cells. The authors should consider to mention and discuss this finding.

We have redone these experiments including some modifications regarding tumor size at day of sacrifice, processing and tissue digestion protocols (please see response to Reviewer 2, query#6), and now find that a sizeable fraction of tumor-infiltrating T cells express CD103 (new Suppl. Fig 6A). Variable expression levels of CD103 in tumor and tissue-resident T cells have been reported before. Casey et al. (1) showed that persistent antigen stimulation decreased CD103 expression by T cells in small intestine IELs. Also, a recent study (2) shows CD103 is expressed on T cells in peritumoral skin but not on tumor-infiltrating T cells in the tumor core. Whereas in our previous experiments we had used mice with 4-week old MC38 tumors, in the new experiments we used tumors that had grown for 3 weeks. Slightly shorter exposure to chronic antigen and smaller tumor sizes, coupled with avoidance of trypsin during tumor digestion in the new experiments, might have all contributed to the significantly higher percentages of intratumor CD69⁺CD103⁺ T cells we found in our new experiments.

Lines 150-151: The authors state that “lymph node T cells... are enriched in non-circulating CD8+ T cells (...average 97.2%)”. How were these identified as non-circulating? Is this based on CD69 expression?

After discussion with an internationally-recognized expert in tissue resident T cells [Dr. Laura Mackay (University of Melbourne)], we have modified the entire section referring to circulating vs. non-circulating/tissue-resident T cells to make it clearer and more accurate. That sentence was considered confusing, as pointed out by the reviewer, and therefore it was removed.

Also, do the lymph nodes drain the region of implanted tumours?

Lymph nodes adjacent and non-adjacent to the region of implanted tumors were pooled to obtain sufficient cells for the experiments. For the sorting experiments, inguinal and axillary lymph nodes were pooled; for tissue-resident survival experiments, only inguinal lymph nodes were used. In all cases, mice had been implanted with a tumor on the right flank. Therefore, in the text we refer to “lymph node T cells” in general, since not only those lymph nodes closest to the tumor were used, although they were included in the sample. Information about which lymph nodes were used in each experiment is now provided in Materials and Methods (p.20).

Fig 6A: What does “%CD8+EYFP+ detected in fragment recipient” on the y-axis refer to? Is this the percentage of whole splenocytes?

That is correct. The axis legend has been modified and now reads “%CD8⁺EYFP⁺ in fragment recipient’s splenocytes”.

Line 122: “CD11c⁺ T cells”. Typo, should be non-T cells, i.e. APC?

Thank you. The typo has been corrected.

Does anti-CD8 Ab treatment ablate tumour-resident T cells – has this been quantified/shown? Alternatively, or in addition, it may act to block CD8 T cells function.

To answer this question we checked the efficacy of anti-CD8 antibodies to deplete intratumoral T cells, since we regularly check depletion only in the peripheral blood. The data shows that CD8⁺ T cells are eliminated from both tumors and blood in mice treated with anti-CD8 antibodies at the dose (200 ug) used in this study, 24 h after treatment. This data has now been included as Supplementary Fig. 1B.

Reviewer #2

This manuscript addresses a very important issue, namely the survival of T cells present within a tumor treated with radiotherapy. Whereas it is known that T cells exhibit different radiosensitivity depending on their maturation and differentiation/activation status, whole body irradiation at relatively low doses has been used to eliminate T cells in clinical scenarios where it is necessary to “create space”, such as in the setting of adoptive T cell therapy. Thus, the commonly held notion is that focal radiation used at therapeutic doses will kill most of the T cells infiltrating a tumor. In the new era of cancer immunotherapy, this has become especially important and a factor influencing the design of clinical studies of radiotherapy and immunotherapy combinations. To address the fate of T cells that are present within the tumor microenvironment at the time of radiotherapy treatment, Arina and colleagues have developed an elegant model based on the T-cell reporter (Lck-EYFP) mouse that enabled them to distinguish between pre-existing and newly primed/recruited T cells following radiotherapy. They show that many T cells survive even after a 20Gy radiation dose, and retain motility and effector functions while losing the ability to proliferate. To explain the differential sensitivity to radiation of tumor-residing T cells, circulating T cells, and T cells residing in lymph nodes and spleen, a number of experiments are performed. The main conclusion that tumor-residing T cells are similar to tissue-resident memory (TRM) T cells is consistent with a growing literature about the presence and role of TRMs in tumors, in both mouse models and patients. However, this conclusion is based on gene expression analysis of bulk sorted CD8 T cells and does not allow the assessment of what percentage of the intratumoral T cells are TRMs. The heterogeneity of the tumor-residing T cells is only superficially investigated, as is the potential influence of other immune cells on T cell survival and functionality in irradiated tumors. Mechanistically, the claim that TGFb is responsible for determining T cell radiosensitivity is not fully supported by the data, as detailed in specific comments below. As mentioned in discussion, TGFb is the target of some immune-oncology strategies, due to its immunosuppressive role. New bifunctional agents (e.g. TRAP, targeting PDL-1 and TGFb) are undergoing clinical testing. Thus, it

is important to determine if this pleiotropic factor plays a role in T cell radiosensitivity, but also critical to provide solid evidence as the stakes are high for the development and optimal use of TGFb-targeting therapies. Finally, tumor-residing T cells are likely to be an important mechanism influencing the response of the tumor to SBRT only in the case of “hot” tumors. For poorly immunogenic mouse tumors, and for the majority of cancer patients, it will be far more important if radiation can generate new T cells. Overall, the data provided about the sensitivity of T cells present within tumors to radiation are relatively convincing and have the potential to advance the field. However, there are several weaknesses on the mechanistic aspects of the study that need to be addressed.

We thank the reviewer for these comments that will be individually addressed below.

Major issues:

1) On page 4 the statement: “A single 200 µg dose of anti-CD8 depleting antibodies on day 0 allowed the tumors to grow unimpeded by any “vaccination effects” induced by SIY-expressing tumor cell inoculation”. This conclusion is not justified. Supplementary Fig 1 shows that in CD8-depleted mice the tumors grow faster and in all mice. However, the large number of T cells present in these tumors in untreated mice, shown in Fig 1 indicates that during tumor growth anti-tumor immunity develops. Presumably CD8 T cell numbers recover quickly after the initial depletion. The immune infiltrate should be characterized at the time of irradiation for relative presence of CD4 and CD8 T cells, Tregs, and their activation/exhaustion state. Some characterization is provided in Supplementary Fig 3, but this is limited to a few markers and to CD8 T cells. Markers of resident memory T cells are not included.

We have performed an additional experiment (new Sup. Fig 1C), to complement those experiments in Sup. Fig 3; together, now we show percentages of CD4, CD8 T cells, Tregs and expression of PD1, CD39, LAG3, TIGIT, 41BB, CD62L, CD44 by CD8⁺ T cells present in the immune infiltrate at the time of irradiation. CD8⁺ T cells indeed recover from initial depletion at the beginning of the experiment, infiltrate the tumor and acquire a phenotype compatible with that of exhausted cells, based on expression of several exhaustion markers, including PD1, CD39, LAG3 and TIGIT. Changes in the main text are on page 4. We have also modified the statement referred to above, such that now it reads: “A single 200 µg dose of anti-CD8 depleting antibodies on day 0 allowed SIY-expressing tumors grow more aggressively and in all mice”. Regarding resident memory markers, we have decided to refer to those later on the manuscript (see also response to query Q#6), to maintain the structure and flow of ideas in the paper.

2) Supplementary Fig 3: Why only PD1 and CD39 were analyzed to assess the effects of irradiation on the endogenous and adoptively transferred CD8 T cells?

Our objective in Suppl. Fig. 3 was to compare the general phenotype of preexisting vs. new tumor-infiltrating T cells, before and after IR. We included PD1 in the flow cytometry staining as the most commonly used marker of exhaustion, and CD39 because it identifies the “most exhausted” T cells (3). Our reasoning was that endogenous T cells possibly had higher levels of markers whose expression increases with longer exposure to

the tumor microenvironment. Other markers included in the panel were ki67 and naïve/memory markers CD44 and CD62L (Fig 3B and 3C).

The similar percentage of H2AX γ foci in the endogenous and adoptively transferred CD8 T cells suggests that most of the adoptively transferred CD8 T cells were already in the tumor at the time of radiation.

Since H2AX γ foci are analyzed almost immediately after IR (usually within 1 h), we would not be able to detect new infiltration in such a short time, and therefore we cannot make that conclusion. The adoptively transferred H2AX γ ⁺ T cells that we see must be cells that have infiltrated within the last 4 days (time since adoptive transfer) and have received radiation 1 h before analysis.

The small difference in CD39 and PD1 expression may simply reflect the fact that the adoptively transferred CD8 T cells all bear the same TCR and recognize the same antigen, whereas the endogenous T cells are a heterogeneous population.

That is possible, and now we offer this as an alternative explanation to our finding (page 5)

3) Supplementary Fig 4A: Why such strange radiation protocol of 10Gy + 20Gy 4 days later is chosen for another tumor model, MC38? Is this clinically relevant?

Although 10+20 Gy is rarely used clinically, the biological equivalent of these doses are used in some SBRT treatment schedules. Our reasoning to use the schedule referred to above was that, in a longitudinal in vivo imaging experiment, we could test increasingly higher IR doses and measure the effects after each dose. In any case, using the accumulated 30 Gy dose in MC38 (Suppl. Fig 4) or 20 Gy single-dose Panc02SIY model (Fig. 1), we reach the same conclusion: a sizeable number of preexisting T cells survive high doses of IR.

Since the data are based only on the detection of EYFP⁺ cells by imaging, can it be ruled out that some of the positivity seen after radiation reflects phagocytosis of dying T cells by myeloid cells rather than surviving T cells? Without more evidence the statement that “at all time points, including those obtained after the 20 Gy dose, tumor resident T cells were detectable” is not fully warranted.

It is unlikely that myeloid cells with engulfed dying EYFP⁺ T cells would make an important contribution to the number of EYFP⁺ cells quantified by us before/after IR for the following reasons:

- we used a Fiji-based macro to quantify the T cells in an automated way. The parameters of this macro were customized to count T cells by limiting counted cells to those with area and circularity parameters expected from lymphocytes moving in tissue spaces. We have now added this information to the Materials and Methods section describing “Quantitative analysis of images”.
- EYFP is well known to be pH and halide sensitive, so phagocyte-ingested T cells would rapidly quench in the phagosome environment. Our images show bright, uniform EYFP distribution that would be expected for cytoplasmic expression, versus the envisioned compartmental dead cell possibility.

- we have performed flow cytometry experiments to directly address the reviewer’s question (**Fig. 1 for Reviewers**).

Fig 1 for Reviewers. Composition of the intratumoral EYFP⁺ cell compartment in T cell reporter mice before/after local 20 Gy treatment. Lck-EYFP mice bearing established (2 week) MC38 tumors were treated as in Sup. Fig 4, with 8 Gy WBI shielding the tumor to eliminate circulating EYFP⁺ T cells while preserving intratumor EYFP⁺ T cells. This was followed by local treatment of the tumor with 20 Gy in some mice and flow cytometric analysis 24 h after local IR. A. Gating strategy is shown from an unirradiated tumor-bearing mouse. B. Summary of data. For obtaining the percentage of total T cells, percentages of CD4⁺ and CD8⁺ T cells as gated on A were added (e.g. 79.9% in the example shown). Data are pooled from two independent experiments with 1 unirradiated and 1-2 irradiated tumor mice each.

In these experiments, we analyzed which immune cell type corresponded with the EYFP⁺ cells detected before or 24 h after radiation (from the time points analyzed in Sup. Fig 3). This time point was selected because phagocytosis is usually measured within a few hours after dying cells become available to phagocytes (4-6). Our flow cytometry experiments show that T cells constitute the vast majority of gated EYFP⁺ events before and after treatment, whereas myeloid cells are only a small fraction.

4) Figure 3A: this figure makes an important point about the radiation resistance of T cells present in the tumor versus the blood. However, data are presented as “% T cells” and it is unclear what the denominator is. The number and not the percentage of T cells should be shown.

As the reviewer notes, the goal of this figure is to highlight the relative resistance of intratumoral compared with circulating T cells. We find this is most clearly shown when using percentages, since percentages allow using the same scale for both compartments and tend to show a smaller variation between individuals and independent experiments than absolute number of cells per gram (tumor) or microliter (blood). However, we agree that the axis label and description were not clear enough. To improve clarity of the figure presentation and the description of methods used to quantify T cell populations, we have included a new Supplementary Figure (Suppl. Fig. 5), that shows the gating strategy and quantification examples. As a reference, other figures in this manuscript show the effect

of IR in intratumoral T cells in terms of absolute T cell number/gram of tissue (e.g. Fig 5C and Suppl. Fig. 6B) after a high dose of IR (8 Gy).

5) *Figure 3B and C: CD4 T cells should be further separated into regulatory and non-Treg, given data that they have a differential radiosensitivity (Kachikwu, E. L., Int J Radiat Oncol Biol Phys 2011).*

In response to this question, we conducted a new experiment to determine the percentage of Treg within the total CD4⁺ T cells found in tumors and peripheral blood after an intermediate (3 Gy) and high (10 Gy) doses of WBI, chosen from the wider range of doses tested in Figure 3. As seen in **Fig. 2 for Reviewers**, we could not detect statistically significant differences in the percentages of Tregs in CD4⁺ T cells from blood or tumors from mice receiving WBI compared with unirradiated mice, at 24 h.

Fig. 2 for Reviewers. Determination of percentages of Tregs in circulating and intratumor CD4 T cells surviving radiation 24 h after WBI. Mice bearing established (2 week) MC38 tumors received 3, 10 Gy or no WBI (N=5 per group). 24 h later, PBL and tumors were stained with antibodies specific for CD3, CD8, CD4 and FoxP3 and analyzed by flow cytometry. Shown is the percent FoxP3⁺ in the CD3⁺CD4⁺ population. CC3⁺CD4⁺ T cells left in peripheral blood after 10 Gy WBI were insufficient to determine %FoxP3⁺ (average 64 cells in the 10 Gy vs. 1429 in the 3 Gy and 7399 in the unirradiated control group). Differences between unirradiated/irradiated groups were not significant (unpaired t test).

This is consistent with different (slower) kinetics of Treg enrichment in blood compared with other organs (e.g., one study found no significant increase in the percent of Tregs in total CD4 in peripheral blood at day 0.5 after WBI with 2 Gy, whereas at day 5 after WBI this increase was significant) (7). In the spleen, however, the increase was already evident 0.5 days after WBI (7). The work cited by this reviewer also analyzed changes in spleen, at day 2 after WBI with 2 Gy (8). Other studies reporting a higher radio-resistance of Tregs in mice looked also in the spleen and later time points (e.g., day 7, (9), day 5-20 (10)). Since changes in Treg representation among IR-surviving CD4⁺ T cells seem to be dose-, organ- and time-dependent, have been published by other groups, and are not the focus of our current study (centered in CD8⁺ T cells, the predominant subset in intratumoral T cells and the one we studied in tissues as well), we opted for simplifying previous Figures 3B-C (new Fig 3B) to only show that most T cells surviving IR in tumors are CD8⁺ T cells. To explicitly state that our study is focused on the CD8⁺ T cell compartment, we have added the following sentence in the main text (p. 6): “The majority of T cells in tumors at any WBI dose were CD8⁺ T cells, in contrast with

circulating CD8⁺ T cells, whose percentage dropped with increasing doses of WBI (**Fig. 3B**), proving the uniqueness of intratumor CD8⁺ T cell survival to IR.”

6) Figure 3D: None of the comparisons between controls and irradiated mice seems significant due to the large variability in controls. Some correction for organ size/weight may reduce the variability. Moreover, the lungs contain several peri-bronchial lymph nodes and bronchial-associated lymphoid tissue, which, like spleen and LN will contain a large fraction of naïve T cells. The latter are known to be more radio-sensitive than memory T cells. In fact, in the lungs there is a marked reduction in CD8 T cells after IR. The variable inclusion of lymph nodes/lymphoid tissue in each of the organs may explain the big variability seen even in the absence of treatment. Overall, this analysis is too superficial to support the conclusion that the “organ microenvironment” determines T cell radiosensitivity. Despite the high CD69 expression, there are very few CD103⁺ cells in the tumor (Supplementary Fig 4B). Thus, another variable is the type of T cells that are present in each organ/tissue site analyzed (e.g., TRM versus T effector memory, etc...). Additional markers, including markers known to define naïve and TRM in different organs, (e.g., LFA1 in liver) would help support a more cautious interpretation of the results.

We discussed the concerns raised by this reviewer with an internationally-recognized expert, Dr. Laura Mackay (University of Melbourne), and performed a new series of experiments. Based on recommendations from Dr. Mackay, we now refer to cells that both exclude intravascular staining antibody and express a well-characterized set of T_{RM} markers as tissue “resident” T cells (e.g., CD103⁺CD69⁺ for IEL, CD69⁺LFA1⁺ for liver). Consequently, we have eliminated the term “tumor-resident T cells” for experiments where we didn’t use T_{RM} markers, and use the more generic “tumor infiltrating” “intratumoral” or “preexisting T cells” instead. CD8⁺ T cells that are found inside tissues and exclude intravascular staining-antibody, including but not limited to T_{RM}, are referred to as “parenchymal” CD8⁺ T cells.

To address other specific concerns listed above, we have:

- Quantified CD8⁺ T cells within tissues as number of cells/gram of tissue
- Taken measures to minimize inter-individual and inter-experimental variability, such as have one investigator perform the same experimental step in all mice and experiments (e.g., mouse perfusion with PBS, IEL processing etc.)
- Used a more gentle tumor digestion protocol that does not require trypsin (whereas we briefly used trypsin after tumor digestion with liberase/DNAse in our original experiments). We have updated our Materials and Methods section accordingly.
- Analyzed parenchymal T cells from different organs for expression of naïve/memory T cell markers
- Decided not to include data from lungs because numbers of T cells were so low after IR that there were not enough cells with a T_{RM} phenotype to analyze.

After adopting these measures, we detected a substantial population of CD69⁺CD103⁺ CD8⁺ T cells in MC38 tumors. As discussed in response to Reviewer 1, comment #3, we used slightly smaller tumors in our new experiments, which could also contribute to the

higher percentages of CD103⁺ T cells we found. Our new experiments suggest that, while in some organs (liver), CD8⁺ T cells with a T_{RM} phenotype seem to specifically be the radioresistant cells among parenchymal CD8 T cells, in other organs such as gut or in the tumor, both T_{RM} marker-expressing cells and those T cells that don't express T_{RM} markers, are similarly radioresistant. Naïve cells are present among parenchymal lymph node T cells but negligible in spleen, while T cells from both organs are similarly radiosensitive; therefore, the presence of naïve T cells cannot explain the differential radiosensitivity of lymphoid vs. non-lymphoid solid organs. Other factors determined by local organ environment might explain the higher radioresistance of IEL and tumor T cells irrespective of their expression of T_{RM} markers. We have modified Fig. 3, Suppl. Fig. 6 and the main text (p.6-7, 12-13) and added Suppl. Table 1 to include this new data.

7) *Page 7: it is not clear what are the data supporting the statement that “Like tumor T cells, lymph node T cells.....are enriched in non-circulating CD8+ cells” in Figure 3D?* Please see our reply to previous query #6: following Dr. Laura Mackay’s advice we now avoid using potentially confusing nomenclature and have therefore removed that sentence.

8) *Supplementary Fig 5: The analysis of the molecular pathways activated by IR should include statistical significance.*
The Y axis lacked a legend. We now added that legend to the Y axis which contains statistical significance values, indicated as $-\log_{10}(\text{P-value})$.

9) *Supplementary Figure 6 lacks a legend to indicate the magnitude of change in expression of the genes in the heat map.*
We have now included that legend that was missing in the first version.

10) *Supplementary Fig 7: The analysis of the molecular pathways activated in tumor vs lymph nodes should include statistical significance analyses. The legend lacks sufficient information to understand what is shown and how the analyses were performed.*
As for query #8, the problem was the lack of label on the Y axis with statistical significance values. We have now added that axis legend. We have also modified the figure legend to more clearly state what is shown in the figure and how the analyses were performed.

11) *On Page 8, the statement “Functional analysis detected TGFβ as the top upstream regulator of T cell reprogramming in the tumor microenvironment” is only supported by z score and p value. It is not clear how this conclusion is supported by data. For instance, metabolic re-programming of T cells in the TME has been reported in several publications, and in Suppl Table 1 Hypoxia ranks a lot higher than TGFβ signaling. Moreover, TGF-β is known to drive CD103 expression in TRM T cells (Nizad et al., Nat Comm 2017) but CD103 was barely expressed by the intratumoral CD8 T cells studied here (Supplementary Fig 4B). In addition to addressing these issues, the pathways that are under control of TGFβ and are selectively activated in the tumor T cells should be shown.*

We have revised our transcriptional analysis with the help of Dr. Sean Pitroda who has expertise in bioinformatics analyses. We now include as Extended Data a full version of the analysis of upstream regulators of T cell reprogramming in the tumor microenvironment by IPA. These data show that TGFb signaling has the highest P value of overlap with the list of genes upregulated in tumor compared to LN T cells (therefore is the top “upstream regulator”), and lists the target molecules downstream of TGFb activation that are present in such upregulated gene list. Former Suppl. Table 1 (new Suppl. Table 2) has been modified to rank hallmark signatures by enrichment score, which is a more appropriate way to present the data. While angiogenesis and EMT rank #1 and 2 and TGFb #12 in this list, all these signatures share a high proportion of activated gene functions (such as growth factors and peptidases) and are reflective of a similar overall tumor-associated biology. Regarding CD103 expression by tumor T cells, please see above response to query #6.

12) Fig 5A shows that the tumor CD8 T cells are more similar to TRM T cells than to spleen T cells, something that is not surprising, and also not very novel.

It is true that several other studies have by now made similar observations, which support our data. This piece of information is important to support the rationale behind testing TGFb as one possible mechanism involved in T cell radioresistance, due to its role in reprogramming tumor T cells (according to our IPA analysis), and in the maintenance of TRM cells in solid tissues, since tumor and TRM share a relatively radioresistant phenotype.

13) Data in Fig. 5C do not demonstrate that blocking TGFb increases radiation sensitivity of the T cells present in the tumor. TGFb was blocked starting 2 days after MC38 inoculation. This led to a large increase in T cells density within the tumor (did it also reduce tumor growth?). Thus, at the time when radiation is used, the tumors in mice treated with TGFb blockade may be different in many other ways from the untreated tumors, and without data to show that the populations of T cells present are phenotypically and functionally similar the conclusion that TGFb controls radiosensitivity is not warranted. In Supplementary Fig 9, the differences do not appear to be statistically significant. Experiments addressing in more depth the role of TGFb signaling on T cell response to radiation are needed.

REDACTED

Overall, this new research is not mature enough yet for publication and will take significantly more time and space than is available now for the current manuscript, which is focused on proving the basic fact that preexisting T cells in tumors treated with IR can resist radiation and contribute to the effect of IR. It is important to publish this finding, so that it can be taken into consideration by clinicians and researchers testing immune/radiotherapy combinations. In response to the reviewer’s concerns, we have added a paragraph discussing other possible factors that could contribute to intratumor T cell radioresistance, and thoroughly revised the text and figure 5/legend to avoid the interpretation that TGFb is the direct and only mechanism behind tumor T cell radioresistance.

14) Supplementary Figure 12B: there is no statistically significant change in ability of CD11c+ cells to stimulate naïve T cells at day 5 post-irradiation. The text should be corrected to recognize this, or additional experiments performed to determine if the CD11c+ DC are more stimulatory. Importantly, it should be shown if DC numbers and phenotype are altered by radiation.

In response to these comments, we have (i) corrected the text to explicitly acknowledge that changes in the ability of CD11c⁺ from irradiated tumors to stimulate T cells were not statistically significant, and (ii) analyzed quantitative/phenotypic changes in the DC compartment after IR (p. 11). Flow cytometry analysis of DC including the distinction between CD103⁺/CD11b⁺ DC was adapted from (17). The new results are shown in Suppl. Fig. 12G-I and suggest that DC are enriched within tumor-infiltrating leukocytes at day 5 after IR. The CD11b⁺ subset seems to be the most relevant in terms of the response to IR based on percentage increase among total DC and higher expression of MHCII after IR.

15) Supplementary Figure 12 E: Changes in MDSC are compared to non-IR tumors. However, over the course of 9 days, much can change also in the absence of treatment as tumors progress. Thus, IR and non-IR tumors should be compared for each of the days shown.

Data for this figure were originally pooled from two experiments using mice bearing well established (more than 3 weeks) Panc02SIYCerlean tumors: in Exp. #1, mice had received local tumor IR 1 or 3 days before sacrifice, or been left untreated; all these mice were sacrificed at the same time; in Exp. #2, mice had received tumor IR 5 or 9 days before sacrifice or been left untreated; all these mice were sacrificed at the same time. Therefore, for each experiment, IR and non-IR tumors were compared and tumors in control mice had grown for as long as tumors in treated mice. Percentages of myeloid cell subsets were compared between the two sets of control mice from the two experiments. and found to be not significantly different, therefore we pooled the control mice We now explain in greater detail how these experiments were performed in the figure legend.

16) Discussion: In the sentence “Here we show that the T cells present in solid tumors at the time of treatment, are not eliminated by radiation doses and schedules typically used in the clinical setting, and appear to be essential to the anti tumor effects of therapeutic IR.” The second part is true only if there are sufficient T cells in the tumor at the time of IR. In many cases in patients this is not the case.

This is correct, and we have now clarified this point by modifying the above referenced opening sentence in the Discussion, which now reads “Here we show that the T cells present in inflamed/immunogenic solid tumors at the time of treatment are not eliminated by radiation doses and schedules typically used in the clinical setting, and appear to be essential to the anti tumor effects of therapeutic IR”. The relative importance of pre-existing vs. newly infiltrating T cells in inflamed vs. non-inflamed human tumors is further discussed starting on line 337, as follows: “Our results show that pre-existing tumor T cells survive IR and mediate antitumor responses. Peripheral newly tumor infiltrating T cells likely also contribute to the anti-tumor effects and are important for at

least two reasons: (...) (ii) The tumor models used here (MC38 and Panc02SIY EGFP) have some baseline T cell infiltration, similar to human “inflamed” tumors. For tumors that have no infiltrating T cells (“non-inflamed”), irradiation could be a way to attract newly infiltrating peripheral T cells and potentially render them into more “inflamed” tumors that could e.g. respond to immunotherapy.”

Minor issues:

1) Page 9, line 220: remove “T” after CD11c+

That typo has been corrected.

2) Fig 1 legend for (D): the explanation of the “N of optical regions” is unclear.

We have modified that sentence to make it clearer.

3) Figure 2 C and E: the different colors used for T cell tracks should be defined as to their meaning

We have added an explanation for the color code used for T cell tracks on the figure legend.

4) Supplementary Fig 3: the flow plots in D are not informative as shown. Gating strategy should be shown.

We now include gating strategy for plots in Fig 3D.

5) Supplementary Fig 3 B and C: define what red and blue colors indicate

Thank you. We have now added a legend containing that information.

1. Casey, K. A., K. A. Fraser, J. M. Schenkel, A. Moran, M. C. Abt, L. K. Beura, P. J. Lucas, D. Artis, E. J. Wherry, K. Hogquist, V. Vezys, and D. Masopust. 2012. Antigen-independent differentiation and maintenance of effector-like resident memory T cells in tissues. *J Immunol* 188: 4866-4875.
2. Park, S. L., A. Buzzai, J. Rautela, J. L. Hor, K. Hochheiser, M. Effern, N. McBain, T. Wagner, J. Edwards, R. McConville, J. S. Wilmott, R. A. Scolyer, T. Tuting, U. Palendira, D. Gyorki, S. N. Mueller, N. D. Huntington, S. Bedoui, M. Holzel, L. K. Mackay, J. Waithman, and T. Gebhardt. 2019. Tissue-resident memory CD8(+) T cells promote melanoma-immune equilibrium in skin. *Nature* 565: 366-371.
3. Gupta, P. K., J. Godec, D. Wolski, E. Adland, K. Yates, K. E. Pauken, C. Cosgrove, C. Ledderose, W. G. Junger, S. C. Robson, E. J. Wherry, G. Alter, P. J. Goulder, P. Klenerman, A. H. Sharpe, G. M. Lauer, and W. N. Haining. 2015. CD39 Expression Identifies Terminally Exhausted CD8+ T Cells. *PLoS Pathog* 11: e1005177.
4. Hu, B., J. Sonstein, P. J. Christensen, A. Punturieri, and J. L. Curtis. 2000. Deficient in vitro and in vivo phagocytosis of apoptotic T cells by resident murine alveolar macrophages. *J Immunol* 165: 2124-2133.
5. Nakayama, M., H. Akiba, K. Takeda, Y. Kojima, M. Hashiguchi, M. Azuma, H. Yagita, and K. Okumura. 2009. Tim-3 mediates phagocytosis of apoptotic cells and cross-presentation. *Blood* 113: 3821-3830.

6. Uderhardt, S., M. Herrmann, O. V. Oskolkova, S. Aschermann, W. Bicker, N. Ipseiz, K. Sarter, B. Frey, T. Rothe, R. Voll, F. Nimmerjahn, V. N. Bochkov, G. Schett, and G. Kronke. 2012. 12/15-lipoxygenase orchestrates the clearance of apoptotic cells and maintains immunologic tolerance. *Immunity* 36: 834-846.
7. Qu, Y., B. Zhang, S. Liu, A. Zhang, T. Wu, and Y. Zhao. 2010. 2-Gy whole-body irradiation significantly alters the balance of CD4+ CD25- T effector cells and CD4+ CD25+ Foxp3+ T regulatory cells in mice. *Cell Mol Immunol* 7: 419-427.
8. Kachikwu, E. L., K. S. Iwamoto, Y. P. Liao, J. J. DeMarco, N. Agazaryan, J. S. Economou, W. H. McBride, and D. Schaeue. 2011. Radiation enhances regulatory T cell representation. *Int J Radiat Oncol Biol Phys* 81: 1128-1135.
9. McFarland, H. I., M. Puig, L. T. Grajkowska, K. Tsuji, J. P. Lee, K. P. Mason, D. Verthelyi, and A. S. Rosenberg. 2012. Regulatory T cells in gamma irradiation-induced immune suppression. *PLoS One* 7: e39092.
10. Zhou, Y., H. Ni, K. Balint, J. K. Sanzari, T. Dentchev, E. S. Diffenderfer, J. M. Wilson, K. A. Cengel, and D. Weissman. 2014. Ionizing radiation selectively reduces skin regulatory T cells and alters immune function. *PLoS One* 9: e100800.
11. Kajita, M., K. N. McClinic, and P. A. Wade. 2004. Aberrant expression of the transcription factors snail and slug alters the response to genotoxic stress. *Mol Cell Biol* 24: 7559-7566.
12. Inoue, A., M. G. Seidel, W. Wu, S. Kamizono, A. A. Ferrando, R. T. Bronson, H. Iwasaki, K. Akashi, A. Morimoto, J. K. Hitzler, T. I. Pestina, C. W. Jackson, R. Tanaka, M. J. Chong, P. J. McKinnon, T. Inukai, G. C. Grosveld, and A. T. Look. 2002. Slug, a highly conserved zinc finger transcriptional repressor, protects hematopoietic progenitor cells from radiation-induced apoptosis in vivo. *Cancer Cell* 2: 279-288.
13. Perez-Losada, J., M. Sanchez-Martin, M. Perez-Caro, P. A. Perez-Mancera, and I. Sanchez-Garcia. 2003. The radioresistance biological function of the SCF/kit signaling pathway is mediated by the zinc-finger transcription factor Slug. *Oncogene* 22: 4205-4211.
14. Choi, J., S. Y. Park, and C. K. Joo. 2007. Transforming growth factor-beta1 represses E-cadherin production via slug expression in lens epithelial cells. *Invest Ophthalmol Vis Sci* 48: 2708-2718.
15. Thuault, S., E. J. Tan, H. Peinado, A. Cano, C. H. Heldin, and A. Moustakas. 2008. HMGA2 and Smads co-regulate SNAIL1 expression during induction of epithelial-to-mesenchymal transition. *J Biol Chem* 283: 33437-33446.
16. Thuault, S., U. Valcourt, M. Petersen, G. Manfioletti, C. H. Heldin, and A. Moustakas. 2006. Transforming growth factor-beta employs HMGA2 to elicit epithelial-mesenchymal transition. *J Cell Biol* 174: 175-183.
17. Broz, M. L., M. Binnewies, B. Boldajipour, A. E. Nelson, J. L. Pollack, D. J. Erle, A. Barczak, M. D. Rosenblum, A. Daud, D. L. Barber, S. Amigorena, L. J. Van't Veer, A. I. Sperling, D. M. Wolf, and M. F. Krummel. 2014. Dissecting the Tumor Myeloid Compartment Reveals Rare Activating Antigen-Presenting Cells Critical for T Cell Immunity. *Cancer Cell* 26: 938.

REVIEWERS' COMMENTS:

Reviewer #1 (Remarks to the Author):

The authors have appropriately addressed all my queries.

Reviewer #2 (Remarks to the Author):

The authors have answered satisfactorily all of my prior concerns. The revised manuscript is improved and the major findings are rigorously demonstrated.